# Biogeography and ecology of Ostracoda in the U.S. northern Bering, Chukchi, and Beaufort Seas

**Laura Gemery**[1,2]*, **Thomas M. Cronin**[1], **Lee W. Cooper**[2], **Harry J. Dowsett**[1], **Jacqueline M. Grebmeier**[2]

**1** U.S. Geological Survey, Florence Bascom Geoscience Center, Reston, VA, United States of America,
**2** Chesapeake Biological Laboratory, University of Maryland Center for Environmental Sciences, Solomons, MD, United States of America

* lgemery@usgs.gov

**Data Availability Statement:** AOD-2020 is available at NOAA's National Centers for Environmental Information (NCEI) World Data Service for Paleoclimatology for download (https://

## Abstract

Ostracoda (bivalved Crustacea) comprise a significant part of the benthic meiofauna in the Pacific-Arctic region, including more than 50 species, many with identifiable ecological tolerances. These species hold potential as useful indicators of past and future ecosystem changes. In this study, we examined benthic ostracodes from nearly 300 surface sediment samples, >34,000 specimens, from three regions—the northern Bering, Chukchi and Beaufort Seas—to establish species' ecology and distribution. Samples were collected during various sampling programs from 1970 through 2018 on the continental shelves at 20 to ~100m water depth. Ordination analyses using species' relative frequencies identified six species, *Normanicythere leioderma*, *Sarsicytheridea bradii*, *Paracyprideis pseudopunctillata*, *Semicytherura complanata*, *Schizocythere ikeyai*, and *Munseyella mananensis*, as having diagnostic habitat ranges in bottom water temperatures, salinities, sediment substrates and/or food sources. Species relative abundances and distributions can be used to infer past bottom environmental conditions in sediment archives for paleo-reconstructions and to characterize potential changes in Pacific-Arctic ecosystems in future sampling studies. Statistical analyses further showed ostracode assemblages grouped by the summer water masses influencing the area. Offshore-to-nearshore transects of samples across different water masses showed that complex water mass characteristics, such as bottom temperature, productivity, as well as sediment texture, influenced the relative frequencies of ostracode species over small spatial scales. On the larger biogeographic scale, synoptic ordination analyses showed dominant species—*N. leioderma* (Bering Sea), *P. pseudopunctillata* (offshore Chukchi and Beaufort Seas), and *S. bradii* (all regions)—remained fairly constant over recent decades. However, during 2013–2018, northern Pacific species *M. mananensis* and *S. ikeyai* increased in abundance by small but significant proportions in the Chukchi Sea region compared to earlier years. It is yet unclear if these assemblage changes signify a meiofaunal response to changing water mass properties and if this trend will continue in the future. Our new ecological data on ostracode species and biogeography suggest these hypotheses can be tested with future benthic monitoring efforts.

www.ncdc.noaa.gov/paleo/study/32312; Cronin et al., 2021). Data obtained during the DBO sampling efforts are available at the Earth Observing Laboratory (EOL) of the National Center for Atmospheric Research (NCAR), http://www.eol.ucar.edu/field_projects/dbo and at the Arctic Data Center (https://arcticdata.io/catalog/portals/DBO), supported by the National Science Foundation.

**Funding:** Financial support for sample collections was provided by grants to JMG and LWC from the NSF Arctic Observing Network program (1204082, 1702456 and 1917469) and the NOAA Arctic Research Program (CINAR 22309.07 and 25984.02, https://arctic.noaa.gov/). Support was also provided by the USGS Climate & Land Use R&D Program. The funders had no role in study design, data collection and analysis, decision to publish, or preparation of the manuscript.

**Competing interests:** Funding for this study, including any from commercial sources, does not alter our adherence to PLOS ONE policies on sharing data and materials.

## Introduction

Biological systems of the Pacific-influenced Arctic Ocean are currently undergoing rapid climate-related transformations [1–4]. This is attributed to ocean warming in the North Pacific, Bering, and Chukchi Seas [5, 6], which is accelerating sea-ice loss and extending the open water season [7, 8]. Changes in sea-ice cover further alter stratification, hydrography, and circulation patterns, all of which influence primary productivity [9–12] and trophic relationships [13, 14]. The quality, quantity, and timing of production reaching the sea floor ultimately affects benthic marine species abundance, distribution, and food web dynamics [2, 3, 15, 16]. In this study, we examine a component of the benthic ecosystem: meiobenthic ostracodes and the primary factors that influence their ecology and biogeography in response to changing conditions in the Bering, Chukchi, and Beaufort Seas.

This study has three primary objectives. The first is to examine large-scale biogeographic ostracode patterns and species diversity in the BCB Seas. The second objective is to identify primary environmental factors related to ostracode faunal distributions, particularly among dominant and diagnostic species. For this, we used ordination analyses of a series of near-shore-to-offshore transects (primarily the DBO lines) to spatially assess species abundance patterns across different water masses. Lastly, we use distribution and abundance patterns of dominant and/or ecologically significant taxa as proxies for species response to environmental changes and consider whether recent temperature, sea ice, and productivity changes are affecting ostracode distributions.

### Ostracodes as indicators of environmental change

Ostracodes are a bivalved group of Crustacea, ranging in size from ~0.5 to 2.0 mm, that secrete a calcareous ($CaCO_3$) shell commonly preserved in sediments. Because individual marine ostracode species have ecological limits controlled by temperature, salinity, oxygen, sea ice, food, and other habitat-related factors, they are useful ecologic indicators [17–20]. Ostracodes provide ways to evaluate the effects of changing environmental conditions in different types of areas/ecotones and through geologic time. In addition to distinct species' ecology, their small size, abundant fossil record, limited stratigraphic ranges, species-level identification, and shell isotope geochemistry, ostracodes are multi-proxy tools for paleoecologic, paleoceanographic reconstructions and biostratigraphy [17–21].

### Controls on marine ecosystems and benthic ostracode species

The distribution of biological communities—from plankton and invertebrates to epibenthic fish and sea birds—has largely been found to reflect the distribution of water masses [16, 22]. While the temperature required for survival and reproduction primarily controls the distribution of cryophilic ostracode species on large biogeographic scales [23, 24], other local factors also influence species spatial dynamics, such as depth, sediment properties, and primary productivity exported to the benthos [17, 25–28]. Assemblage composition in any given area is constrained by a combination of these environmental gradients. In addition to water mass properties, this study examines other parameters that affect dominant and indicator ostracode species over south to north gradients (Fig 1), as well as nearshore to offshore transects (Figs 2 and 3), to better understand the ecological preferences of important indicator species.

Here, we extend a prior study analyzing ostracode distributions in the Bering Sea [31], using additional samples collected through the Distributed Biological Observatory (DBO). This international hydrographic and biological sampling program, initiated in 2010, targets known areas of high biological activity along latitudinal transects in the

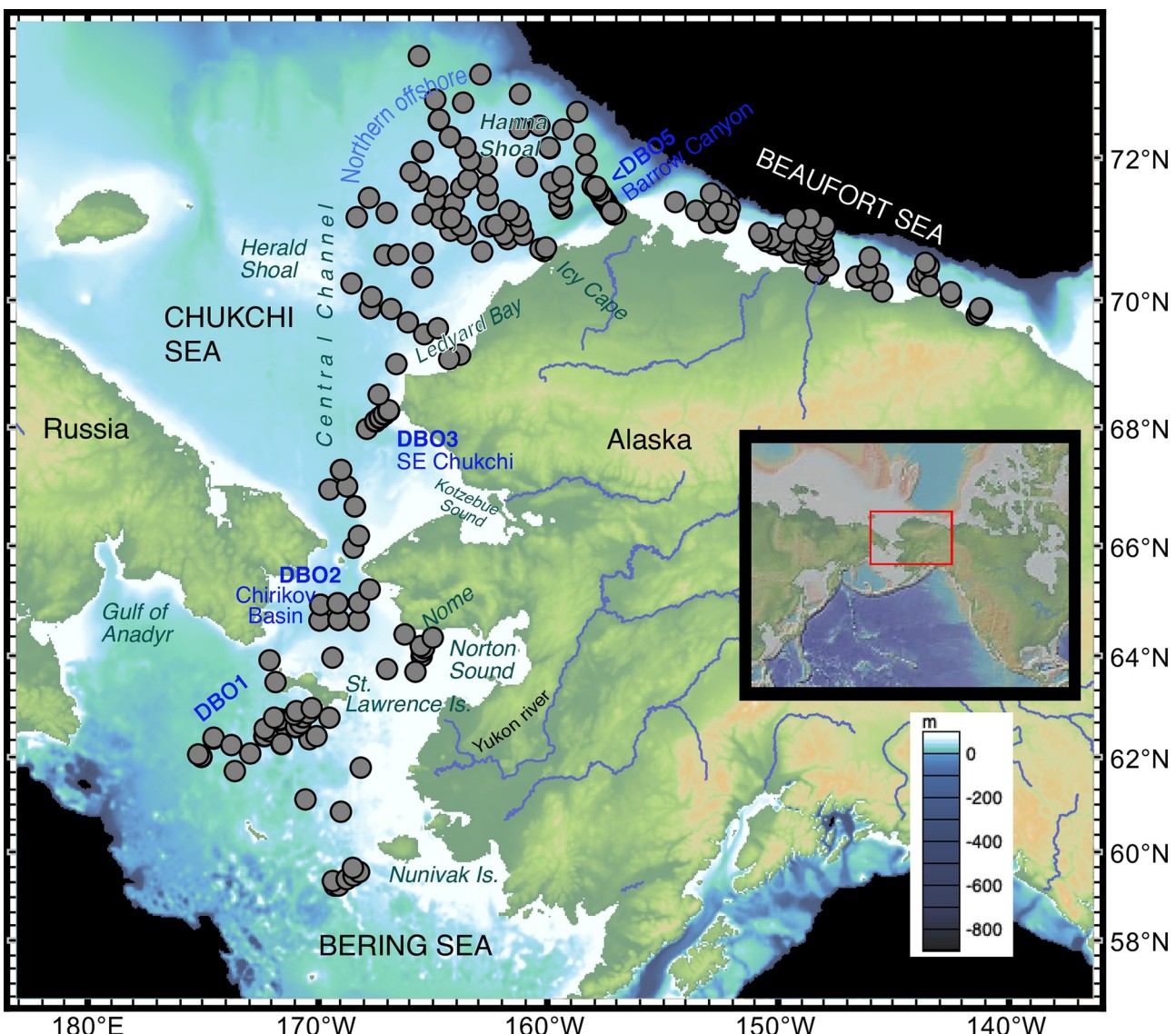

**Fig 1. Locations of samples used for the biogeographical study (n = 289 sediment aliquots with ≥30 specimens per sediment sample, 34,369 total specimens, 1970–2018).** The blue "DBO" areas denote Distributed Biological Observatory stations. (Geographic coordinates are provided in Table 1.) Note that some stations along established DBO transects are not represented due to low or no ostracode abundance (i.e. DBO3-6 to 3–8 and DBO4). Smaller inset map shows context of the study area. Figure made with GeoMapApp [29] (www.geomapapp.org) / CC BY / CC BY [30].

Pacific-Arctic for repeated sampling and ecosystem monitoring [32]. We incorporated an ostracode species analysis into the DBO framework with samples collected from 1970 to 2018 and others in the vicinity of these locations (Fig 1) using a subset of data (S1 Table) from the Arctic Ostracode Database (AOD-2020); [33–36]. The AOD-2020 compiles data for 96 extant ostracode species from 1574 Arctic and subarctic surface (most from the upper 0-2cm) samples, with associated data that include geographic coordinates, water depth, temperature, and salinity. This allowed multivariate analyses to be done on a subset of samples in the northern Bering, Chukchi, and Beaufort (BCB) Seas over the past several decades.

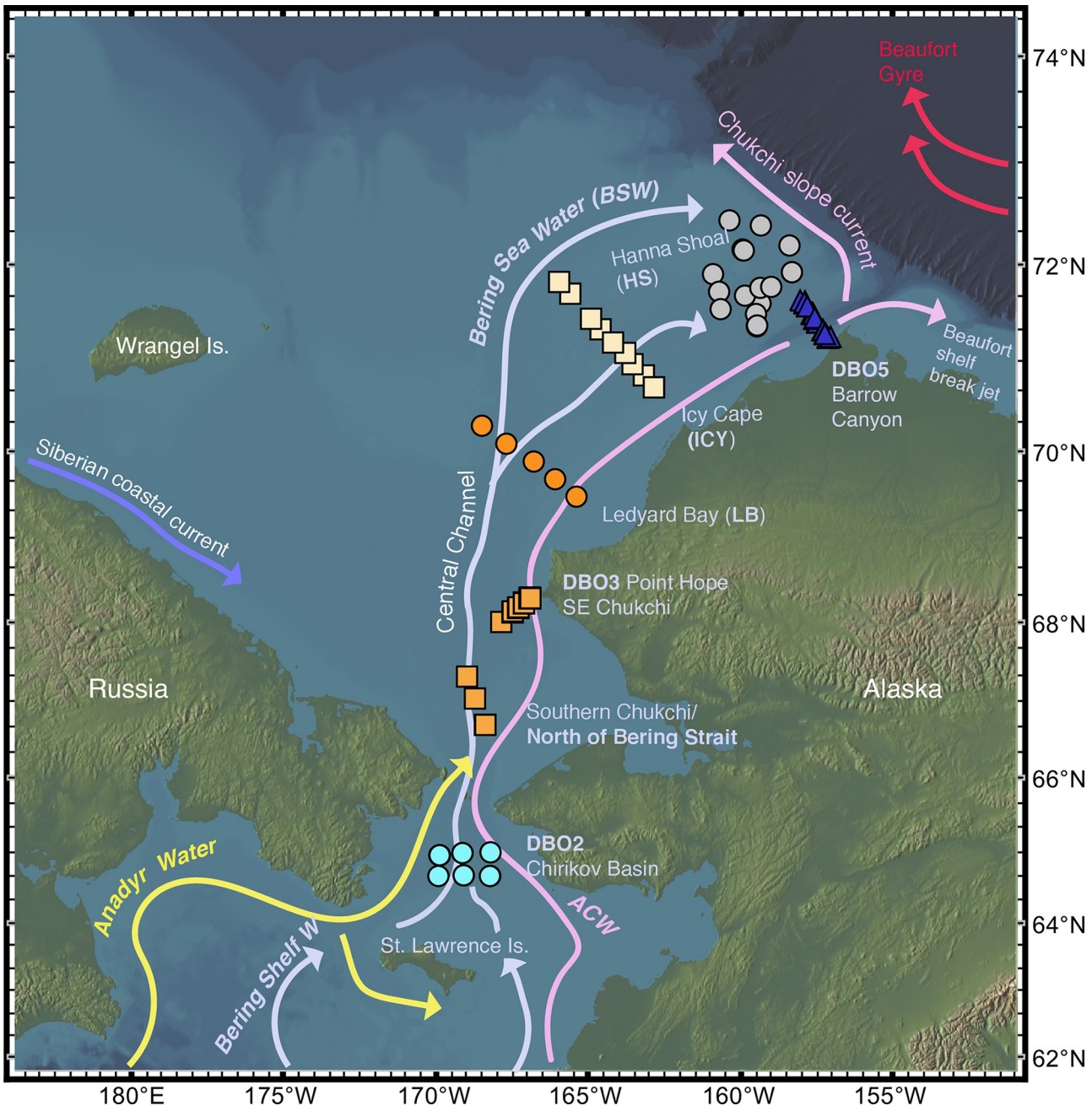

**Fig 2. Location of sample sites in the northern Bering and Chukchi Seas used for ecoregion transect (nearshore to offshore) study (n = 147 surface sediment aliquots with ≥30 specimens per sediment sample, 19,300 total specimens, 1990–2018).** Schematic of summertime surface current flow patterns and water mass type that can affect seafloor conditions (Anadyr Water [AW], Alaska Coastal Water [ACW], Bering Sea Water [BSW]; properties denoted in Table 3) adapted from Danielson et al. (2017, 2020). Figure made with GeoMapApp [29] (www.geomapapp.org) / CC BY / CC BY [30].

## Environmental setting

The continental shelves in the northern BCB Seas include several water masses with different origins that differ in properties, including temperature, salinity, nutrients [37]. The Bering Sea is the major source of nutrients, heat, and freshwater flowing into the Chukchi Sea through the Bering

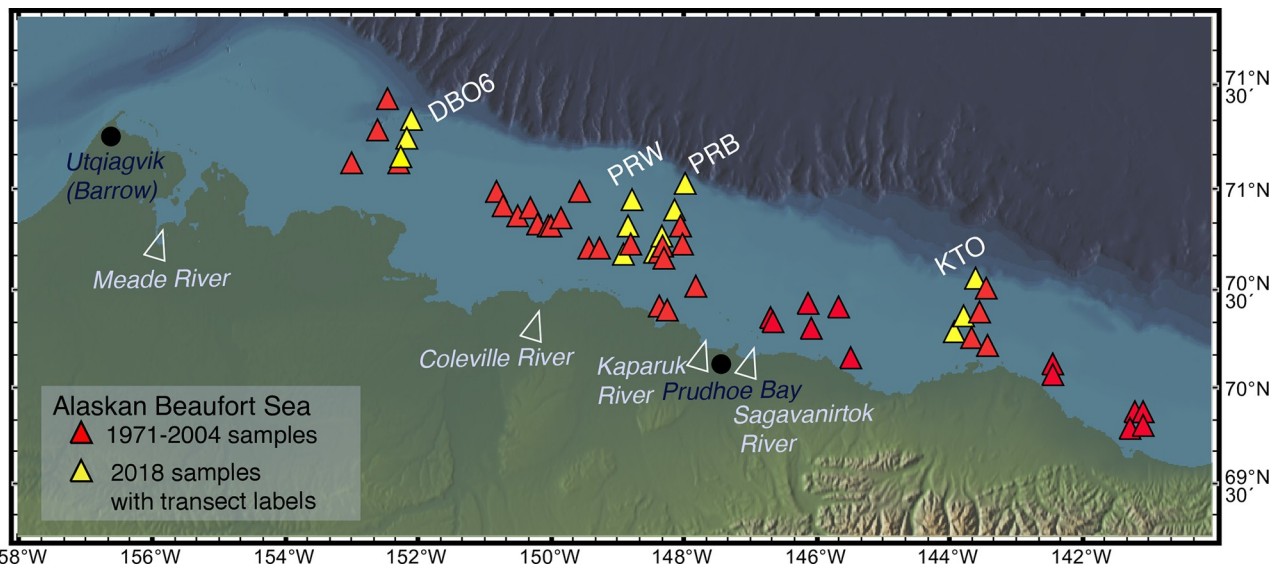

**Fig 3. Location of samples used in Beaufort Sea ecological transect study from years 1971–2018 (n = 55 surface sediment aliquots with ≥30 specimens per sediment sample, 20-100mwd, 6,617 total specimens).** Figure made with GeoMapApp [29] (www.geomapapp.org) / CC BY / CC BY [30].

Strait [6], including three distinct summer water masses: Anadyr Water, Alaska Coastal Water, and Bering Sea Water (Fig 2); [37–39]. Anadyr Water (AW; -1.0˚<T<1.5˚, S≥32.5) is relatively saline, cold, and nutrient-rich and flows along the Russian coast and the western side of the northern Bering Sea. Alaska Coastal Water (ACW; 3˚<T<9˚, S≤32) in summer contains warmer and fresher water derived in part from Alaskan river runoff [38]. Bering Shelf Water (BSW; 0˚< T<3.0˚, 31.8<S<33) is an intermediate water type that is lower in macro-nutrients but with higher salinity contributions from the Bering Slope Current [37]. It runs northward and bifurcates at St. Lawrence Island to enter the Bering Strait between the westerly AW and easterly ACW.

In summer, currents flowing though the Bering Strait drive oceanographic properties in the Chukchi Sea, where water is made up of ACW and Bering Sea Water (BSW), which is a combination of both Bering Shelf Water and AW [39–41]. Bering Sea Water has a higher salinity and nutrient levels than ACW [41, 42]. In the central Chukchi basin, BSW contains the highest primary production and chlorophyll standing stock [43]. This water mass follows bathymetric contours of the Chukchi Sea shelf, flowing northwest to Herald Canyon and northeast across the mid-shelf Central Channel to flank Hanna Shoal to the northern tip of Alaska (Barrow Canyon; Fig 2); [39, 43]. Year-round, when the prevailing northeast winds are weak, outflowing water from the Chukchi Sea is advected toward the eastern Canadian Basin along the Beaufort shelfbreak jet (Fig 2); [44].

These seasonal ocean currents are key in transporting nutrients and phytodetritus into the western Arctic Ocean and maintaining high productivity and biomass [15]. In winter and early spring, the shelf areas are uniformly comprised of cold Winter Water (WW; T≤ -1.0˚, S>31.5); [45, 46]. In summer and fall, WW remains present in the bottom waters of the northern Chukchi Sea [46].

## Materials and methods

### Sample collection and processing

The AOD-2020 compiles ostracode census data collected from circum-Arctic and subarctic sites between 1970 to 2018 from a number of programs and expeditions. Geographical

coordinates, depth, temperature, salinity, and original source methodology for all samples in the database are based on original published studies. Most samples were obtained from the uppermost 0–2 cm of surface sediments. For collections during 2009–2018, including the Distributed Biological Observatory (DBO) program, surface sediments were taken from the tops of multicores or Van Veen grabs prior to opening. Most sampling occurred during the months of June, July, August, or September. Corresponding bottom water temperature and salinity were collected at the time of sampling or if shipboard expedition data were not available, as was the case for some older Beaufort Sea samples from the 1970s and 1980s, measurements were derived from World Ocean Atlas 2001 [47, 48] and World Ocean Database 2013 [35, 49]. Salinity is given without units using the Practical Salinity Scale [50].

Sediments ranging in wet weight from 20 to 150g were washed with tap water though a 63μm sieve, oven-dried in paper filters at 40–50˚C, and then dry sieved over a mesh-size of >125μm. In most cases, all ostracode specimens present in a sample were picked. Living and dead specimens were counted together; however, the majority of samples contained well-preserved shells and carapaces with chitinous appendages that indicated specimens were alive at or near the time of collection. The number of specimens refers to valves and articulated carapaces each counted as one specimen [33].

## Surface sediment characteristics

Because sediment composition and organic carbon content impact benthic faunal biofacies in the Pacific-Arctic [3, 51], we used ancillary, corresponding surface sediment data, where available, to help characterize benthic ostracode habitat in the northern Bering and Chukchi Seas. Surface sediment total organic carbon (TOC), total organic nitrogen (TON), carbon to nitrogen ratios (C/N), sediment chlorophyll-a (sedchla, as a function of newly settled organic matter), and sediment grain size (0–4 phi = coarse pebbles to sands, ≥5 phi = fine silts and clay) used in this study have been previously reported (see Cooper et al., 2002, 2015 [52, 53]; Cooper and Grebmeier, 2018 and methodology therein [54]) and are available in public data archives (PacMARS EOL data portal, http://pacmars.eol.ucar.edu/ [55]; Grebmeier and Cooper, 2014b, 2016 [56, 57]; and Arctic Data Center, https://arcticdata.io/ [58]).

## Statistical analyses

Multivariate statistical analyses were carried out using the Paleontological Statistics (PAST) software package, version 3.24 [59]. Relative frequencies (percent abundance of total assemblage) were calculated for each ostracode species. Species relative abundance data are expressed as a percentage of an individual species relative to the total number of individuals in a sample. Our studies focused on ostracode samples yielding more than 30 total specimens. Both species (R-mode) and samples (Q-mode) were grouped by several different correspondence analyses. A principal component analysis (PCA) was used to group ostracode species by their similarities/differences in dominance based on samples designated by ecoregions. A detrended correspondence analysis (DCA) described relationships between ostracode assemblage structure and geographic DBO/other ecoregions. A canonical correspondence analysis (CCA) was used to explore multivariate relationships between the most common ostracode species and their associations with major environmental variables, and sample locations were color-coded by ecoregion. The benthic environment at each station in the northern Bering and Chukchi Seas was characterized by eight variables that could affect ostracode habitat and survival: bottom water temperature; bottom salinity; grain sizes 0–4 phi; grain size ≥5 phi; sediment chlorophyll-a; TOC; TON; C/N ratios. The Shannon Weaver "H" Index was used to measure ostracode diversity [60]. The values of "H" typically range from 0.0 to 5.0, and

increases with increases in both richness (or the number of different species) and evenness (how close in abundance each species is in an environment). In general, values above 3.0 suggest a habitat structure that stable and balanced. Values less than 1.0 indicate a more challenging or impoverished habitat for survival and reproduction.

## Results

Benthic ostracodes were abundant and diverse at continental shelf depths (20-~100m water depth) in Pacific Arctic surface sediments analyzed (ranging from 30 to 562 specimens/sample; Fig 1, Table 1). A total of 47 ostracode species and five genera were identified in 289 samples (34,369 total specimens) collected from 1970–2018 in the BCB Seas (Table 2). Ostracode species identified represent a mixture of Arctic, subarctic, and cold-temperate taxa. At the sampling sites, summer bottom water temperatures were highly variable, ranging from 12˚C off Nome to 6–9˚C near the Alaskan coast to ≤0˚C in Bering and Chukchi Sea middle shelf areas (Fig 4A–4E). More specifically, ecoregion comparisons of summer near-seafloor temperatures from conductivity-temperature-depth (CTD) measurements show the northern Bering Sea, southwest of St. Lawrence Island (Fig 4A, in year 2018), the Chirikov Basin (Fig 4B, red dots = DBO2 eastern-most samples off Alaska) and nearshore regions in the southeastern Chukchi Sea (Fig 4C, red dots = DBO3-1 eastern samples off Pt. Hope) recently warmed by 2–4˚C above historical temperatures. Summer bottom water temperatures at sample locations in Anadyr Water in the northern Bering Sea (Fig 4B, blue diamonds) and those more centrally located in Chukchi Sea ecoregions (Fig 4C, orange squares = DBO3-8 and Fig 4D, navy squares = DBO5-10) have

**Table 1. Ecoregion names and respective bounding coordinates of sample groups and transects investigated in this study.**

| Ecoregions | DBO1—South and west of St. Lawrence Island, Bering Sea | DBO2—Chirikov Basin, Northern Bering Sea | DBO3—Southern Chukchi Sea | Ledyard Bay (LB) transect, Chukchi Sea | Icy Cape (ICY) transect, Chukchi Sea | C-NE -Northeast Chukchi Sea | DBO5—Barrow Canyon, Chukchi Sea | C-HS—Hanna Shoal, Chukchi Sea | Alaskan Beaufort Shelf |
|---|---|---|---|---|---|---|---|---|---|
| Latitude˚N | 63.77 | 65.111 | 68.60 | 70.30 | 71.83 | 71.00 | 71.808 | 73.30 | 70.60 |
| ˚S | 61.84 | 64.482 | 66.752 | 69.50 | 70.72 | 69.00 | 71.111 | 71.37 | 71.70 |
| Longitude˚E | -172.18 | -167.86 | -166.48 | -165.40 | -162.86 | -159.40 | -155.93 | -158.30 | -131.90 |
| ˚W | -176.14 | -170.49 | -171.41 | -168.50 | -165.97 | -173.80 | -158.84 | -165.60 | -140.00 |
| Depth range (m) | 32–66 | 38–50 | 30–51 | 35–50 | 43–46 | 31–55 | 41–131 | 37–60* | 20–90 |
| Mean summer bottom temp. (˚C) | -1.64 ±0.26 | 2.01 ±2.37 | 2.20 ±1.64 | 8.7 to 2.4 | 1.4 to -0.4 | -0.70 ±1.54 | 0.1±2.3 | -1.7 | -0.9 ±0.7 |
| Mean bottom salinity | 32.47 ±0.32 | 32.2 ±0.61 | 32.36 ±0.56 | 31.7 ±0.53 | 32.1 ±0.1 | 32.59 ±0.84 | 32.6±0.9 | 33.0 ±0.3 | 31.4 ±1.0 |
| Sediment chla (mg m-2) | 13.31 ± 6.51 | 15.74 ± 8.15 | 19.16 ± 9.80 | 6.1 to 20.4 | 6.2 to 26.6 | 12.74 ± 7.73 | 5.2 to 38.6 | 8.7 ±4.1 | not avail |

Samples were grouped by ecoregions based on established Distributed Biological Observatory (DBO) station grids, which include four biological hotspot locations in the Bering and Chukchi Seas along a latitudinal gradient and sampled interannually: DBO1-south of St. Lawrence Island polynya (SLIP); DBO2-Chirikov Basin, north of St. Lawrence Island; DBO3-southern Chukchi Sea, southwest of Point Hope; and DBO5-Barrow Canyon in the northeast Chukchi Sea. Two other transects, sampled only in 2018, are also examined: Ledyard Bay (LB) and Icy Cape (ICY). Two other areas are defined in this study: northeast Chukchi (C-NE) that included sample locations in the northeast not specifically part of DBO4 or DBO5 transects; and Hanna Shoal (C-HS), an offshore region around Hanna Shoal. The symbol "*" represents that a few sample locations are located in deeper water (72, 104 and 130m) in Hanna Shoal. The Alaskan Beaufort Sea was treated as one ecoregion. DBO station mean bottom water temperature, salinity, and sediment chlorophyll, with ± standard deviation, are provided for years 2000–2012, from March to July at DBO1, May to August at DBO2, July to Sept at DBO3, and May to Sept. at Northeast Chukchi. From Grebmeier et al., 2015a [15]; Grebmeier and Cooper, 2014a [61] and Okkonen, 2013 [62]. Ranges for sediment chlorophyll-a are provided for transects LB, ICY and DBO5 from the present study because values at a given location were very divergent from neighboring stations.

**Table 2. List of 47 species and five genera found in the study region (≥1% of cumulative assemblage and/or occurred in 5 or more samples).**

| | |
|---|---|
| A-dun | *Acanthocythereis dunelmensis* (Norman 1869) sensu lato |
| Argil | *Argilloecia* spp. |
| C-clu | *Cluthia cluthae* (Brady, Crosskey & Robertson 1874) |
| C-tesh | *Cytheretta teshekpukensis* Swain 1963 |
| C-macch | *Cytheromorpha macchesneyi* (Brady and Crosskey 1871) |
| C-angul | *Cytheropteron angulatum* Brady and Robertson 1872 |
| C-arcti | *Cytheropteron arcticum* Neale and Howe 1973 |
| C-arcu | *Cytheropteron arcuatum* Brady, Crosskey & Robertson 1874 |
| C-champ | *Cytheropteron champlainum* Cronin 1981 |
| C-elaeni | *Cytheropteron elaeni* Cronin 1989 |
| C-excav | *Cytheropteron excavoalatum* Whatley and Masson 1979 |
| C-inflat | *Cytheropteron inflatum Brady*, Crosskey & Robertson 1874 |
| C-montro | *Cytheropteron montrosiense* Brady, Crosskey & Robertson 1874 |
| C-nodos | *Cytheropteron nodosoalatum* Neale and Howe 1973 |
| C-perlaria | *Cytheropteron perlaria* Hao 1988 |
| C-sulense | *Cytheropteron sulense* Lev 1972 |
| C-suzdal | *Cytheropteron suzdalsky*i Lev 1972 |
| C- tumefact | *Cytheropteron tumefactum* Lev 1972 |
| Cyth-spp | *Cytheropteron* spp. |
| E-concin | *Elofsonella concinna* (Jones 1857) |
| E-neo | *Elofsonella neoconcinna* Bassiouni 1965 |
| F-angu | *Finmarchinella angulata* (Sars 1866) |
| F-barentz | *Finmarchinella barentzovoensis* (Mandelstam 1957) |
| F-logani | *Finmarchinella logani* (Brady and Crosskey 1871) |
| H-emarg | *Hemicythere emarginata* (Sars 1866) |
| H-clathrata | *Hemicytherura clathrata* (Sars 1866) |
| H-fascis | *Heterocyprideis fascis* (Brady and Norman 1889) |
| H-sorb | *Heterocyprideis sorbyana* (Jones 1857) |
| Jonesia | *Jonesia acuminata* (Sars 1866) sensu lato |
| K-arcto | *Kotoracythere arctoborealis* Schornikov and Zenina 2006 |
| K-hunti | *Krithe hunti* Yasuhara, Stepanova, Okahashi, Cronin and Brouwers 2014 |
| Loxo | *Loxoconcha venepidermoidea* Swain 1963 |
| Munsey | *Munseyella mananensis* Hazel and Valentine 1969 |
| Norman | *Normanicythere leioderma* (Norman 1869) |
| P-limi | *Palmenella limicola* (Norman 1865) |
| P-pseudo | *Paracyprideis pseudopunctillata* Swain 1963 |
| Paracytherois | *Paracytherois* spp. |
| P-chaun | *Pteroloxa chaunensis* Schornikov and Zenina 2006 |
| R-mirab | *Rabilimis mirabilis* (Brady 1868) |
| R-septen | *Rabilimis septentrionalis* (Brady 1866) |
| Roberts | *Robertsonites tuberculatus* (Sars 1866) |
| Roundstonia | *Roundstonia globulifera* (Brady 1868) |
| S-bradi | *Sarsicytheridea bradii* (Norman 1865) |
| S-macro | *Sarsicytheridea macrolaminata* (Elofson 1939) |
| S-punct | *Sarsicytheridea punctillata* (Brady 1865) |
| Schizo | *Schizocythere ikeyai* Tsukagoshi and Briggs 1998 |
| Sclero | *Sclerochilus* spp. |

*(Continued)*

**Table 2.** (Continued)

| | |
|---|---|
| S-affinis | *Semicytherura affinis* (Sars 1866) |
| S-complan | *Semicytherura complanata* (Brady, Crosskey & Robertson 1874) |
| S-main | *Semicytherura mainensis* (Hazel and Valentine 1969) |
| Semi-spp | *Semicytherura* spp. |
| S-undata | *Semicytherura undata* (Sars 1866) |

Abbreviations of species are used in stack plots. According to the International Commission on Zoological Nomenclature, authors' names are in parentheses if the assignment listed was not the first description of the species.

not yet manifested a sustained warming trend [16]. The Chukchi Ecosystem Observatory (CEO) mooring, located 70 miles offshore in northeast Chukchi Sea near Hanna Shoal, (71.6˚N and 161.5˚W) rarely recorded near-seafloor temperatures above 0˚C prior to 2016. That year and each following year it recorded several months with temperatures exceeding 0˚C, which lasted up to four months in 2018 [1]. Bottom water temperatures representing a similar but wider geographic area in 2015 and 2017 corroborate this finding (Fig 4E).

Salinity was relatively uniform across the sampling locations in that marine conditions prevailed and averaged 32 ±2. In the Alaskan Beaufort Sea (20-90m water depth), summer temperatures across all years of sampling averaged 0˚C ±1˚C, and salinity averaged 31±1. Coarser, sandy sediments that occur in the hydrodynamically active nearshore areas did not cause lower ostracode abundances, as might have been expected [63, 64].

Diversity "H" in the Chukchi Sea ranged from 1.8 to a little more than 2 during most sample years. Samples averaged 12 (±4) different species. The Beaufort Sea samples had comparable diversity values of 1.8, which was consistent in years 1970–2018 where sampling occurred, and 11 (±3) distinct species per sample. Ostracode diversity in the northern Bering Sea was the lowest compared to these regions, and averaged 1.1–1.2 during the period 2009–2018 with an average of 6 (±2) species per sample.

## Large-scale biogeography of ostracodes in the Bering, Chukchi, Beaufort Seas

Three dominant species in BCB Seas region accounted for 20% to 50% of the total population in the study area: *Normanicythere leioderma*, *Sarsicytheridea bradii*, and *Paracyprideis pseudopunctillata* (Fig 5). Each of these species clearly groups with sampling regions designated by DBO ecoregions (noted by symbol on legend) in separate quadrants of the PCA (Fig 5). In some samples along the DBO3 and central Chukchi Sea (Ledyard Bay) transects (indicated by orange squares on the PCA), *N. leioderma* and *S. bradii* were co-dominant. Otherwise, all three taxa showed distinct spatial distribution patterns related to the path of the primary summer bottom water masses (and its respective properties) in the surveyed regions. Other ostracode species (labeled in blue around the center axes on the PCA) are secondary components in assemblages.

A DCA (Fig 6A) grouped ostracode samples in four distinct geographical groups (Table 3). These ostracode biofacies are associated with specific bottom water masses, sediment substrates and/or food sources: Group 1 is a subarctic assemblage dominated by *N. leioderma* in the vicinity of the St. Lawrence Island polynya and the western Chirikov Basin at DBO2 that is influenced in summer by Anadyr Water. Group 2 is an Arctic-subarctic assemblage dominated by *S. bradii* and *N. leioderma* found in the eastern Chirikov Basin of DBO2 and nearshore areas in the southeast Chukchi Sea. Group 2 and Group 3 is bridged by *S. bradii*. In sediments further offshore in the central and middle shelf areas, *P. pseudopunctillata* was the most

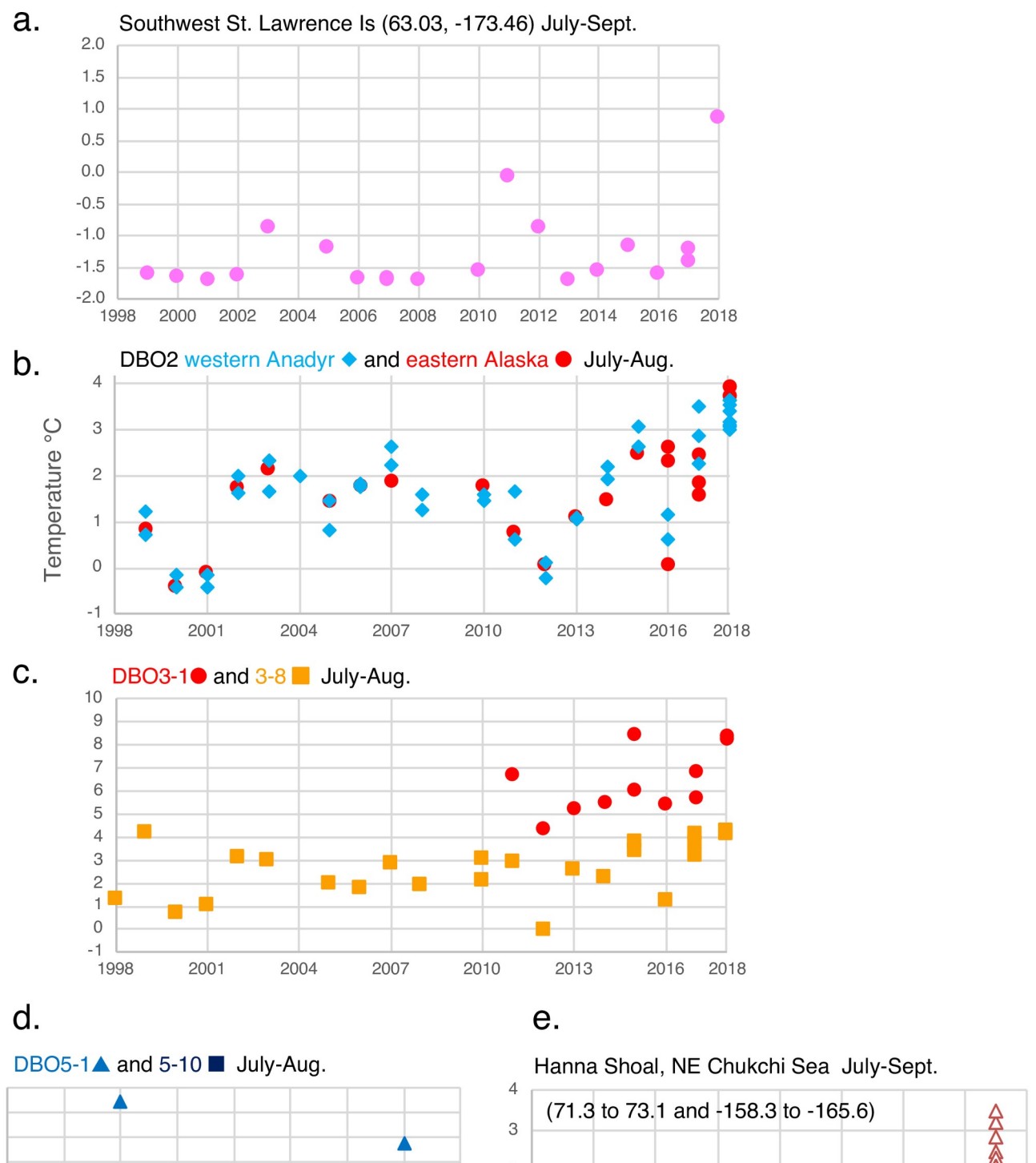

**Fig 4. Summer bottom water temperature at ecoregions.** Comparison of summer bottom temperatures in studied ecoregions and months at stations nearest to shore (e.g. DBO3-1, DBO5-1) that are locations affected by Alaska Coastal Water, and more central or middle-shelf stations (DBO3-5 or DBO5-10) that are affected by Bering Sea Water (winter or summer variants). This subset of data was derived from the Pacific Marine Arctic Regional Synthesis (PacMARS) Project (Grebmeier and Cooper, 2014a [61]; http://dx.doi.org/10.5065/D6VM49BM; Okkonen, 2013 [62]; https://data.eol.ucar.edu/dataset/10339) and recent expedition data (2014–2018) taken aboard USCGS Healy and CCGS Sir Wilfrid Laurier and archived at the National Science Foundation's Arctic Data Center [58].

abundant species in Group 3, influenced by the BSW of the northeast Chukchi Sea. This offshore sample group includes sediments of Hanna Shoal, which is characterized by stable, cold bottom water temperatures and higher salinity from brine rejection, with WW present in bottom waters year-round [46]. Group 4 are the Beaufort Sea samples where influences by river influx and sea ice are strong, and ostracodes are dominated by *P. pseudopunctillata* and *H. sorbyana*.

Environmental factors influencing 14 of the most abundant ostracode species in the northern Bering and Chukchi Seas were examined by CCA (Fig 6B) and indicate that variables other than temperature and salinity water mass characteristics influence species distribution patterns. The environmental variables best correlated to ostracode assemblage structure at nearshore stations in the Chukchi Sea (orange squares in Fig 6B, i.e. DBO3-1 to 3–4, Ledyard Bay, and blue triangles at DBO5-1 to 5–3) are coarse, sandy sediments (0-4phi) and increased temperatures of ACW. The species most associated with these characteristics include *Schizocythere ikeyai*, *Munseyella mananensis*, *Semicytherura mainensis*, and *Elofsonella concinna*. In addition, *N. leioderma* and *S. bradii* are associated with these regions where sediments have sandy-pebbly textures and also at some locations with high sediment chlorophyll-a in the northern Bering Sea, both south of St. Lawrence Island and in the Chirikov Basin. In samples

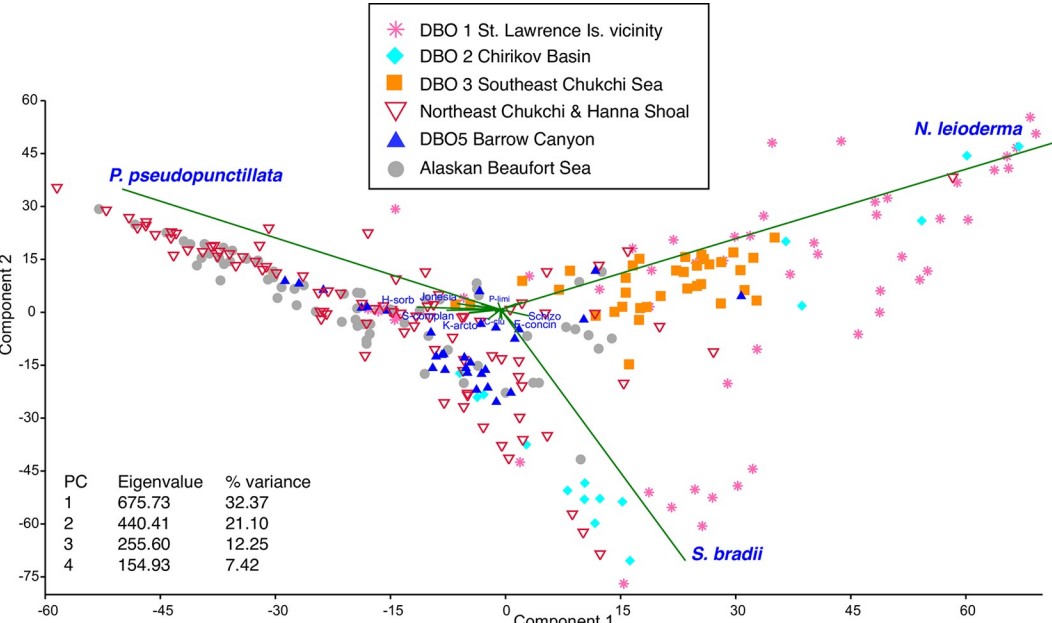

**Fig 5. Principal Correspondence Analysis (PCA).** PCA of ostracode assemblages in the northern Bering, Chukchi and Beaufort Sea region (locations on Fig 1; geographic coordinates in Table 1), with sites designated by DBO areas and other ecoregions (legend symbols) and major taxa (green lines with species names labeled in blue). Three species dominate the assemblage groups: *N. leioderma*, *S. bradii, and P. pseudopunctillata*. The proximity of the symbols in the plot reflects the level of similarity in taxa present at each sampling station.

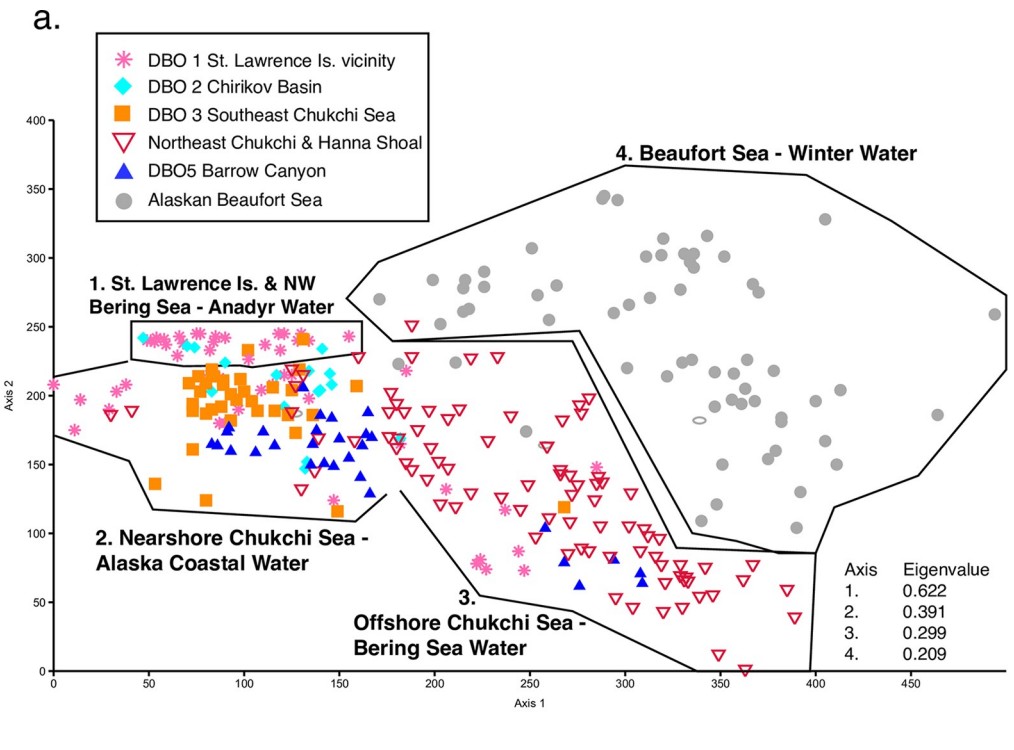

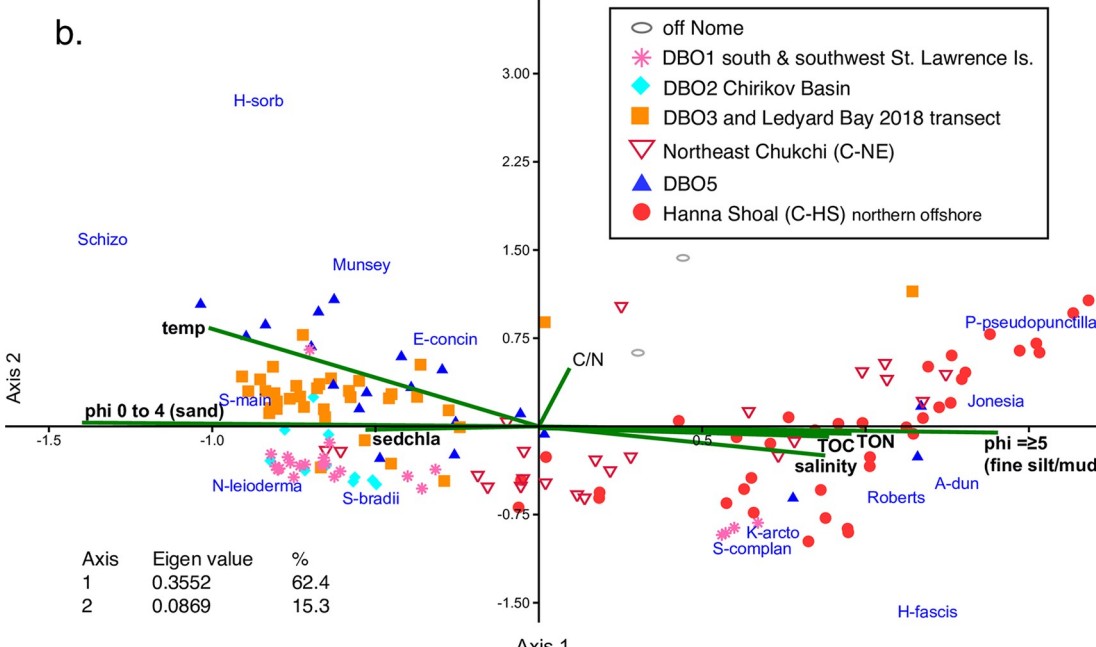

**Fig 6. a.** Detrended correspondence analysis (DCA). DCA using the northern Bering, Chukchi and Beaufort Seas biogeographic dataset (1970–2018), describes relationships between ostracode assemblage structure and geographic ecoregions in the region. Ostracode biofacies differed among ecoregions within the study area (described in Table 3) and grouped primarily on summer water mass characteristics (temperature and salinity). **b.** Canonical correspondence analysis (CCA). CCA of ostracode assemblages in the northern Bering-Chukchi Sea region, with sampling ecoregion sites (noted by symbol on legend), major taxa (names are labeled in blue), and eight environmental parameters (green lines and labels in black: temperature, salinity, sediment total organic carbon [TOC], sediment nitrogen [TON], sediment chlorophyll-a [sedchla], phi ≥5 [fine silt], and phi 0–4 [gravel pebbles to sands]). Samples in the Hanna Shoal region (red dots) were separated from the larger northeastern Chukchi Sea sample set (inverted red triangle). Dataset includes the 14 most abundant species in 153 surface sediment aliquots (23,939 total specimens) from the northern Bering and Chukchi Seas during summers 1990–2018 for which sediment data was available.

**Table 3. DCA results based on the relative abundance of 49 ostracode species (including three genera, *Argilloecia*, *Cytheropteron* and *Semicytherura*) in 289 surface sediment samples from continental shelf regions of the Bering Chukchi and Beaufort Seas collected primarily in summer 1970–2018.**

| Group | Ecoregion | Distinguishing features | Typical bottom summer water mass* | Dominant species | Associated taxa in assemblage |
|---|---|---|---|---|---|
| 1 | Northern Bering Sea—Western Chirikov Basin (DBO2), St. Lawrence Isl polyna | Subarctic-Arctic, polyna open water, high sediment chlorphyll-a, hot spot areas of high-quality production, gravel-sandy sediments | Anadyr (-1.0 to 1.5°C, ≥32.5 salinity) | *Normanicythere leioderma* | *Sarsicytheridea bradii, Palmenella limicola, Semicytherura mainensis, Semicytherura affinis* |
| 2 | Nearshore Chukchi Sea (DBO3-1 to 3–5 and DBO5-1 to 5–7) | Cold-temperate, subarctic, arctic; gravel to sandy sediments | Alaska Coastal Water (~ 3–9°C, ≥30 salinity) | *Normanicythere leioderma* and *Sarsicytheridea bradii* | *Schizocythere ikeyai, Munseyella mananensis, Semicytherura* spp. (i.e. *S. undata*), *Finmarchinella* spp., *Heterocyprideis sorbyana, Cytheropteron nodosalatum, Elofsonella concinna* |
| 3 | Eastern Chirikov Basin (DBO2) and central Bering Sea (middle shelf LB, ICY stations) | Arctic-subarctic; normal marine salinity | Bering Sea Water (0–3°C, 31.8–33.6 salinity) | *Sarsicytheridea bradii* | *Elofsonella concinna, Semicytherura complanata, Kotoracythere arctoborealis, Semicytherura affinis* |
| 3 | Offshore NE Chukchi and Hanna Shoal DBO5-9 and 5–10 | Arctic, marine euryhaline, hot spot areas of high TOC detritus, fine-grained sediments | Bering Sea Water/ Winter Water (≤-1.0°C, ≤31.5 salinity avg. 33 salinity) | *Paracyprideis pseudopunctillata* | *Semicytherura complanata, Heterocyprideis fascis, Robertsonites tuberculatus, Acanthocythereis dunelmensis, Cluthia cluthae, Argilloecia* spp., *Kotoracythere arctoborealis, Jonesia acuminata* |
| 4 | Alaskan Beaufort Sea | Arctic, marine euryhaline, river influenced | Bering Sea Water/ Winter Water (≤-1.0°C, ≤31.5 salinity avg. 33 salinity) | *Paracyprideis pseudopunctillata* and *Heterocyprideis sorbyana* | *Sarsicytheridea bradii, Sarsicytheridea punctillata, Cytheropteron sulense, Heterocyprideis fascis, Acanthocythereis dunelmensis, Robertsonites tuberculatus* |

Four assemblages related to average summer bottom water properties, and several species' preferences of bottom sediment texture, organic sediment food types, are established. Because this area is a broad continental shelf, there is overlap of taxa between Alaska Coastal Water and Bering Sea Water, which is bridged by *S. bradii* and secondary taxa that include *E. concinna*.

located in the offshore northern Chukchi Sea and Hanna Shoal region (red inverted triangles and red circles respectively in Fig 6B), sustained cold temperatures (<0°C), high salinity from brine rejection during ice formation, and finer grained sediments best align to ostracode assemblage structure and the dominant species, *P. pseudopunctillata*, *Jonesia acuminata*, *Acanthocythereis dunelmensis*, *Robertsonites tuberculatus*, *Kotoracythere arctoborealis*, and *Heterocyprideis fascis*.

The CCA yielded four axes that explained 84.4% of the variance in the relationships between ostracode assemblage structure and environmental properties. The first two axes alone explained about 52% of the variance in the data. The CCA affirms that in the northern Bering and Chukchi Sea region, continental shelf ostracode taxa are more correlated to variables associated with distance from shore and water mass than to latitude (60–71°N). These factors include sediment substrate and food type or availability which link certain species to specific environments.

Based on ordination and multivariate results of ostracode species in the study region (Figs 5 and 6B), we consider six species, *N. leioderma*, *S. bradii*, *P. pseudopunctillata*, *S. complanata*, *S. ikeyai*, and *M. mananensis*, to be most diagnostic of specific ranges in bottom water temperatures, salinities, sediment substrates and/or food sources (Figs 5, 6A and 6B; S2 Table; S1 and S2 Figs).

## Ecoregions and transects: Ecological-scale ostracode assemblages

We also incorporated DBO transect lines (Figs 2 and 3) in each ecoregion (Table 1) into our analysis of ecological preferences of dominant and ecologically significant species. Ecoregion-scale plots of species abundances across seven transects over multiple sampling years help illustrate the regional variation of ostracode biofacies and their spatial composition from nearshore to offshore (Figs 7–13). Ecoregion DBO-2 in the Chirikov Basin of the northern Bering Sea (Fig 7) shows a sharp faunal boundary between *N. leioderma* overlain by Anadyr Water on the western side of the Basin and *S. bradii* in Bering Shelf Water on the eastern side. The southeast DBO-3 ecoregion is dominated by *N. leioderma*, *S. bradii* and *S. ikeyai* in ACW. In samples directly north of the Bering Strait in BSW, *S. complanata* increases in proportion among these species (Fig 8). More complex onshore-to-offshore assemblage changes are reflected in the Ledyard Bay and Icy Cape transects (Figs 9 and 10) from 2018, both in the Chukchi Sea. Ledyard Bay and Ice Cape also show clear onshore-to-offshore changes in dominant species. With the exception of a few samples, the Hanna Shoal in the northeast Chukchi Sea ecoregion (Fig 11) is dominated by *P. pseudopunctillata* and Group 3 species (Table 3). The Barrow Canyon transect (Fig 12) shows perhaps the most complex faunal patterns reflecting the temporally and spatially complex and variable ocean water masses and depth changes in the Canyon. The Beaufort Sea west-to-east transect (Fig 13, sample locations on Fig 4 map) shows fairly consistent dominance of *H. sorbyana*, *P. pseudopunctillata*, *S. bradii* and recent contributions from *N. leioderma*, which is in sharp contrast to its abundant presence in samples of the western Chirikov Basin and southernmost Chukchi Sea. Biogeographic patterns of these species are further discussed below in the "Ecological transects" section.

## Time-series analysis

Two time periods (1970–2012 and 2013–2018) were assessed by PCA to evaluate possible temporal changes in ostracode assemblages (Fig 14). These groupings were chosen based on the documentation of accelerated environmental changes [1, 16] during the latter years, including sea ice duration [7], ocean temperature and oceanography [6, 46] and primary productivity [65]. We did not find that the three most dominant species, *N. leioderma* (Bering Sea), *P. pseudopunctillata* (offshore Chukchi and Beaufort Seas) and *S. bradii* (all regions), changed in relative abundance. For the more recent 2013–2018 period, northern Pacific species *Munseyella mananensis* and *Schizocythere ikeyai* exhibited small but ecologically significant increases in abundance (~2%) and frequency in the Chukchi Sea region. Based on this evidence, we examined times series of relative frequencies of these five species, *N. leioderma*, *S. bradii*, *P. pseudopunctillata*, *S. ikeyai* and *M. mananensis* (Fig 15). The goal was to augment the synoptic time slice analyses to determine if the sampling was complete enough to detect decadal trends from interannual variability. The time series was limited to the last 18 years in Chukchi Sea (2000–2018) because too few repeat observations at similar locations to assess benthic change regionally.

## Discussion

### Distribution and ecology of dominant and indicator taxa

*Normanicythere leioderma* and *S. bradii* are ubiquitous in this region, with very broad environmental tolerances. These species are adapted to the large seasonal variations of the inner and middle continental shelf. *S. bradii* is a wide-ranging, cold temperate to frigid species with temperature tolerances of -1.7 to 18˚C [23], but it demonstrates a preference for mid-shelf depths in temperatures colder than 4.5˚C and typical Arctic marine salinities on continental shelves

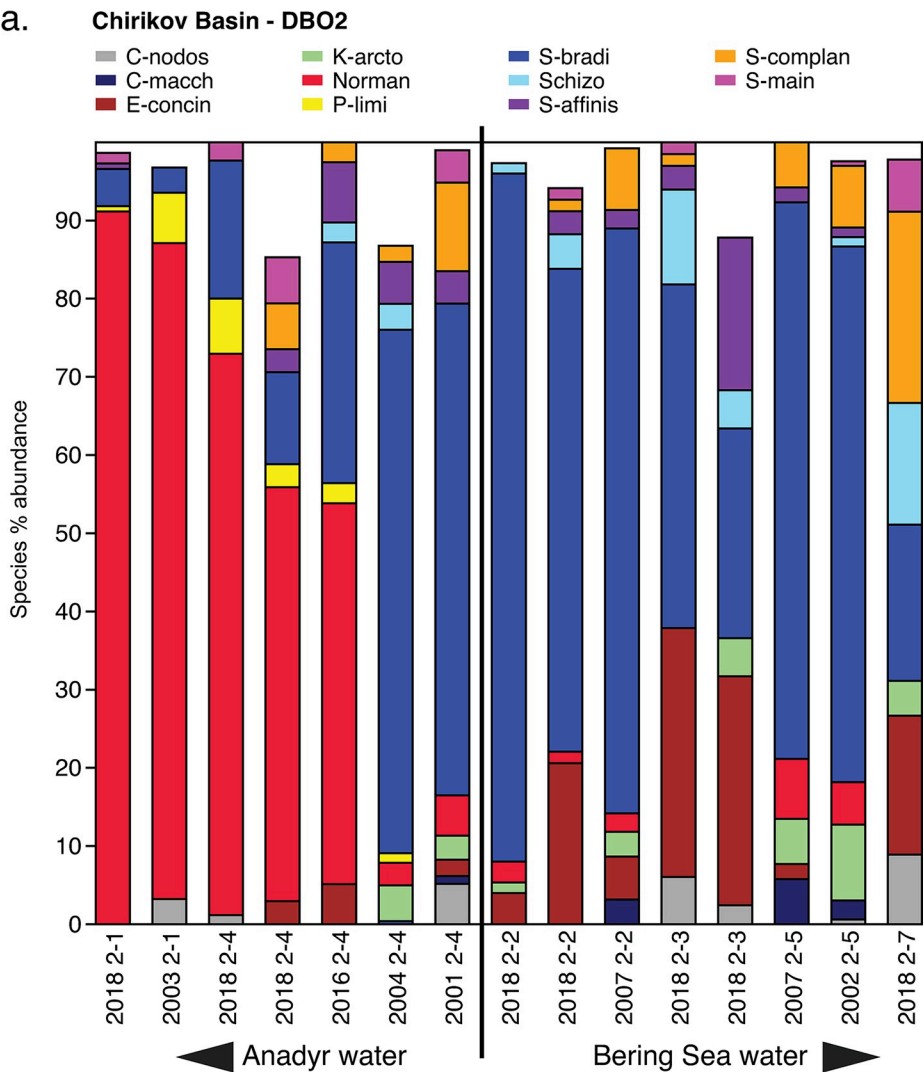

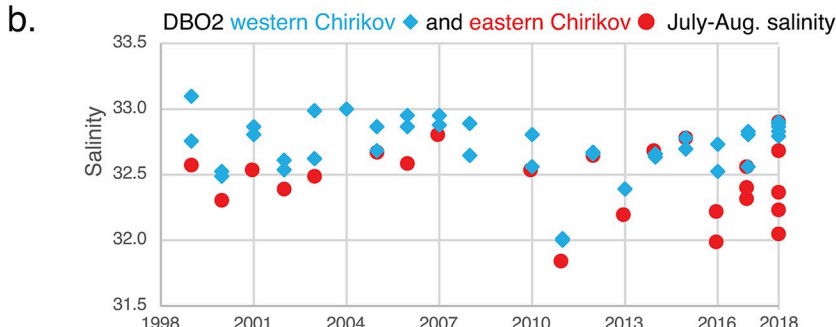

**Fig 7.** a. Primary species, DBO2—Chirikov Basin. Stack plot of dominant species in DBO2- Chirikov Basin region (Fig 2) in the northern Bering Sea (various years from 2001 to 2018; n = 15, 1,314 specimens with ≥30 specimens/ sample). The 11 species presented in the stack plot comprise 98% of the biofacies in this ecoregion. (Note the boundary between Anadyr water on the western side of Chirikov Basin and Bering Sea Water on the eastern side is generalized in this graphic.). b. Summer bottom water salinity at western and eastern DBO2—Chirikov Basin sampling sites. Time series plot of summer bottom salinity at sampling stations on the western and eastern side of the Chirikov Basin (Sources: Grebmeier and Cooper, 2014a [61]; Okkonen, 2013 [62]; and recent expedition data (2014–2018) taken aboard USCGS Healy and CCGS Sir Wilfrid Laurier archived at the Arctic Data Center [58]).

(31–33 is typical; [36]; S1A and S1B Fig). Likewise, *N. leioderma* has wide temperature tolerances (-1.7 to 19˚C; [23]) in typical Arctic shelf salinity (S1A and S1B Fig). As a dominant species in the northern Bering Sea [31], its distribution is circum-Arctic, including the Chukchi, Beaufort, and Eastern Siberian Seas [36], but is very rare in the Kara Sea and absent in the Laptev Sea [20]. *Paracyprideis pseudopunctillata* is a cryophilic, littoral-sublittoral species (1-50m) with a southerly range limit of ~63˚N latitude, tolerant of salinities as low as 5–10 [19, 23, 66, 67]. *S. complanata* is adapted to the coldest conditions on continental shelves, including where polynyas form during winter [67]. An analysis of this species in AOD-2020 reveals that its distribution is tied to near-freezing water temperatures ≤0˚C in sublittoral depths, even though it can withstand higher seasonal temperatures.

These four species, which dominate the study area, are ideally adapted to bottom water temperatures that normally range from −1.8˚C to ~4˚C across the area (S2 Table). This temperature range coincides with the temperature window established for benthic macrofaunal animals prevalent at hotspot areas of the Pacific Arctic [15, 51].

*S. ikeyai* and *M. mananensis* are members of genera endemic to the Pacific that have modern distributions off eastern Japan and the Okhotsk Sea and are adapted to a wider range of summer bottom water temperatures (0–20˚C; [27, 68]) than the species discussed above. *M. mananensis* is also common to the cold temperate North Atlantic with bottom water temperatures from 2˚ to 14˚C and water depths from 24 to 261m [69]. Siddiqui and Grigg [70] included this species in a list of sublittoral fauna from Halifax Harbour, Nova Scotia. Using the AOD-2020 to assess the subarctic-Arctic distribution of these species in samples with 10 or more specimens, *M. mananensis* is most commonly found in estuarine inlets, with highest abundances (2–30%) in Hudson Bay, North Star Bay (Thule, Greenland), Norton Sound (Alaska) and Chaunskaya Gulf (Eastern Siberian Sea) in bottom water temperatures ranging in summer from 0 to 7˚C and salinities from 26 to 33. *S. ikeyai* is rare (one sample reported) in the Beaufort Sea before 2018. It is recorded on the Chukchi Sea shelf in 8 samples (abundance of 2–38%) before 2014 in summer bottom water temperatures from 0˚C to 6˚C and salinities from 31 to 33. Single specimens occurred in three samples in the Alaskan Beaufort Sea from 1971 to 1982. In the northern Bering Sea, *S. ikeyai* is rare (1.2% of the cumulative assemblage in samples collected during 2010–2018), and *M. mananensis* is extremely rare (<1% of assemblage). Increased frequency of these species in the Arctic could reflect warming bottom water temperatures since these species reach peak abundances in lower-latitude waters that generally remain above 0˚C throughout most of the year. Overall, the abundance and distribution of these six species can be used to infer benthic environmental conditions and potential change in the Pacific Arctic in future studies and in sediment archives for paleo-reconstructions.

While temperature of the bottom water mass is a dominant control on ostracode distributions (Fig 6A), the CCA (Fig 6B) showed that some species are correlated to a particular sediment texture, and type/availability of carbon food sources. For a few species, *N. leioderma* and *P. pseudopunctillata*, these factors appear to drive their abundance, as supported by the transect analyses (see "Ecoregion transects" Discussion). For example, in the northern Bering and Chukchi Seas, *N. leioderma* more commonly inhabits coarse and pebbly sediments (0–4 phi) of faster moving water, as does *S. bradii*, *S. ikeyai*, and *M. mananensis* (Fig 6B; S1C Fig). *Paracyprideis pseudopunctillata* is found in greater proportions in silty, fine-grained seafloor habitats (≥5 phi) as indicated by its low abundance in sediments of phi 0–4 (Fig 6B; S1C Fig). *S. complanata* does not demonstrate a sediment preference (S1C Fig).

Availability of organic carbon is a top-level factor that drives species survival and reproduction [71]. *N. leioderma* is associated with areas of newly settled organic material (i.e. higher quality carbon), as indicated by its relative abundance in areas of higher sediment chlorophyll-a (>10mg/m$^2$) and lower C/N ratios (<7; Fig 6B; S2A and S2B Fig). In contrast, *P.*

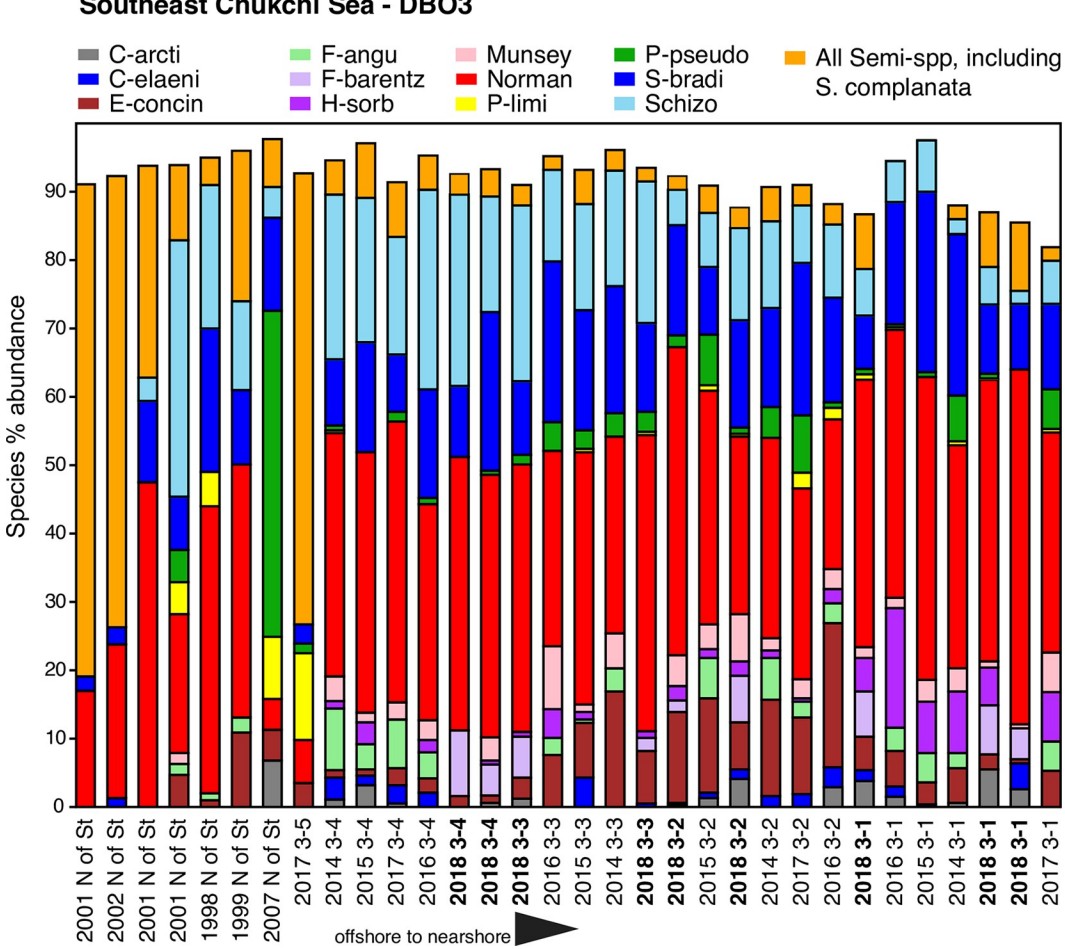

**Fig 8. Primary species, DBO3—Southern Chukchi Sea.** Stack plot of the 12 most abundant species and one genus (*Semicytherura*), comprising 92% of the ostracode assemblage, in DBO3 region (sites on Fig 2 map) from years ranging from 1998 to 2018, (n = 32 samples with ≥30 specimens/sample, 7,514 total specimens).

*pseudopunctillata* resides in areas with lower sediment chlorophyll-a and higher TOC (>0.5%; indicating a greater amount of detritus in the surface sediments; Fig 6B; S2B Fig) and C/N values (>7; indicating greater terrigenous inputs; Fig 6B; S2C Fig). This may indicate *P. pseudopunctillata* uses detritus or more refractory carbon sources for food. The widespread abundance of *S. bradii* in areas of high- and low-quality carbon further supports its eurytopic nature, and it demonstrates no clear carbon (food) preference (S2A–S2C Fig). Because *S. complanata*, *M. mananensis* and *S. ikeyai* are secondary components in assemblages, their proportions are lower and therefore their environmental preferences are more difficult to evaluate. *M. mananensis* and *S. ikeyai* are present in sediments with higher C/N ratios, which corresponds to their higher proportions at sampling stations nearer to shore in ACW, e.g. southern Chukchi Sea stations DBO3-1 to DBO3-4 and northern Chukchi Sea stations DBO5-1 to DBO5-3 (S2C Fig). In these nearshore sediments, coarse grains and gravel may dominant, but there is fine sediment in between the large grain size. This indicates variable settling rates over various seasons, providing complexity to nearshore statistical findings. Additional sampling of these species and corresponding sediments are required to better evaluate the role that sediment and organic carbon sources may have on their abundance.

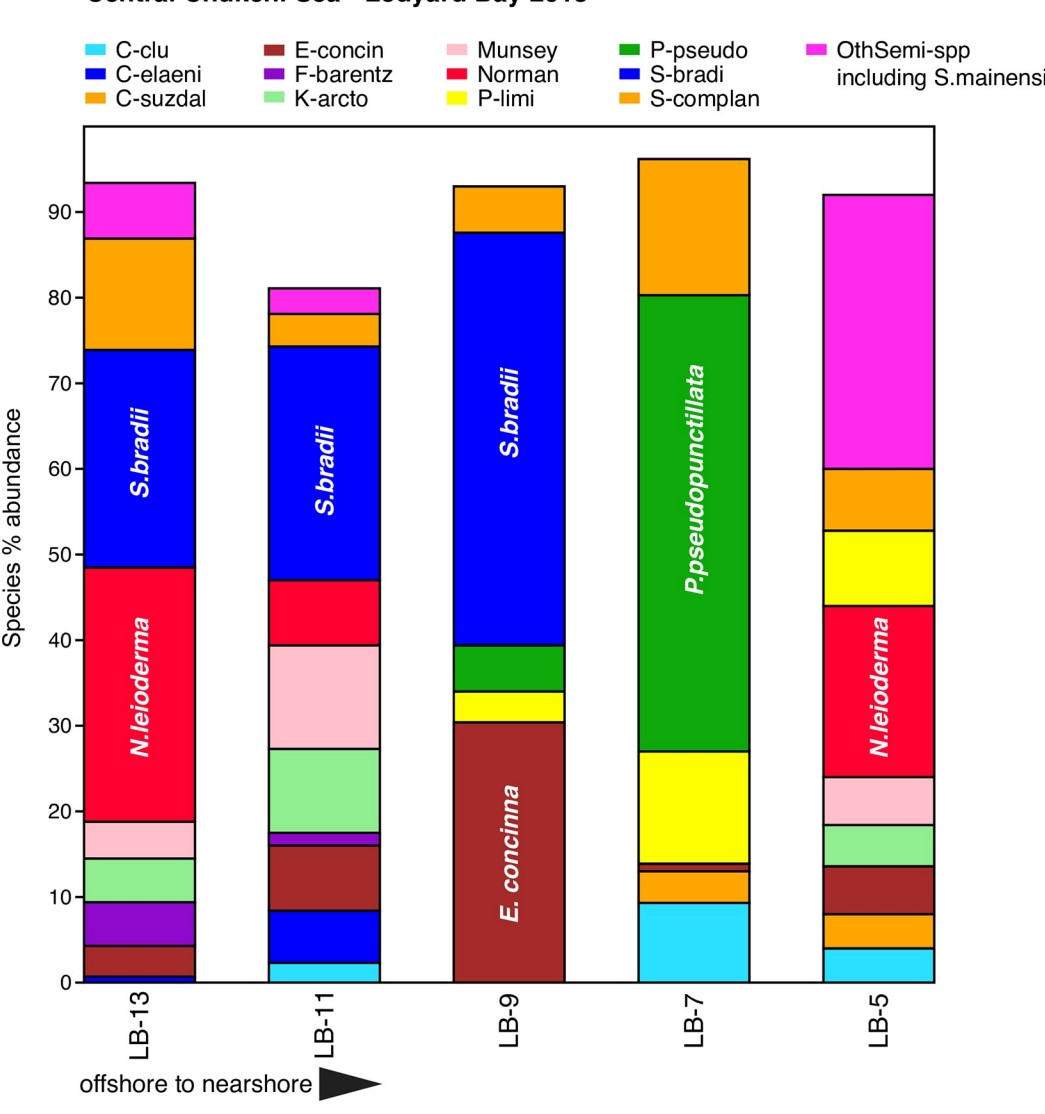

**Fig 9. Primary species, Central Chukchi Sea–Ledyard Bay.** Stack plot of the 12 most abundant species and one genus (*Semicytherura*), comprising >90% of the ostracode assemblage at the 2018 Ledyard Bay transect (sites on Fig 2 map), southeast Chukchi Sea, (n = 5 surface sediment samples, all with ≥50 specimens/sample, 558 total specimens).

## Ecoregion transects: Spatial relationships among ostracode biofacies

Stack plots showing the primary species in sample assemblages in an ecoregion and/or along a sampling transect (Figs 7–13) provide insight into factors that control species occurrence and frequency. Generally in the study area, bottom water temperature declines with increasing latitude and distance from shore. Sediment composition also becomes finer-grained (greater silt fraction) with increasing distance from shore [15, 72]. Other properties, especially in the central and northeast Chukchi Sea, involve more complex spatial patterns [51, 73]. Varying seafloor bathymetry, sediment substrate, flux of organic carbon settling to the seafloor, and current velocities combine with water mass characteristics to create a complex habitat of varying ecological gradients. These transects show the variability of species occupying a sampling area of 0.1m$^2$ at a given location on the seafloor during a summer's day collection. At

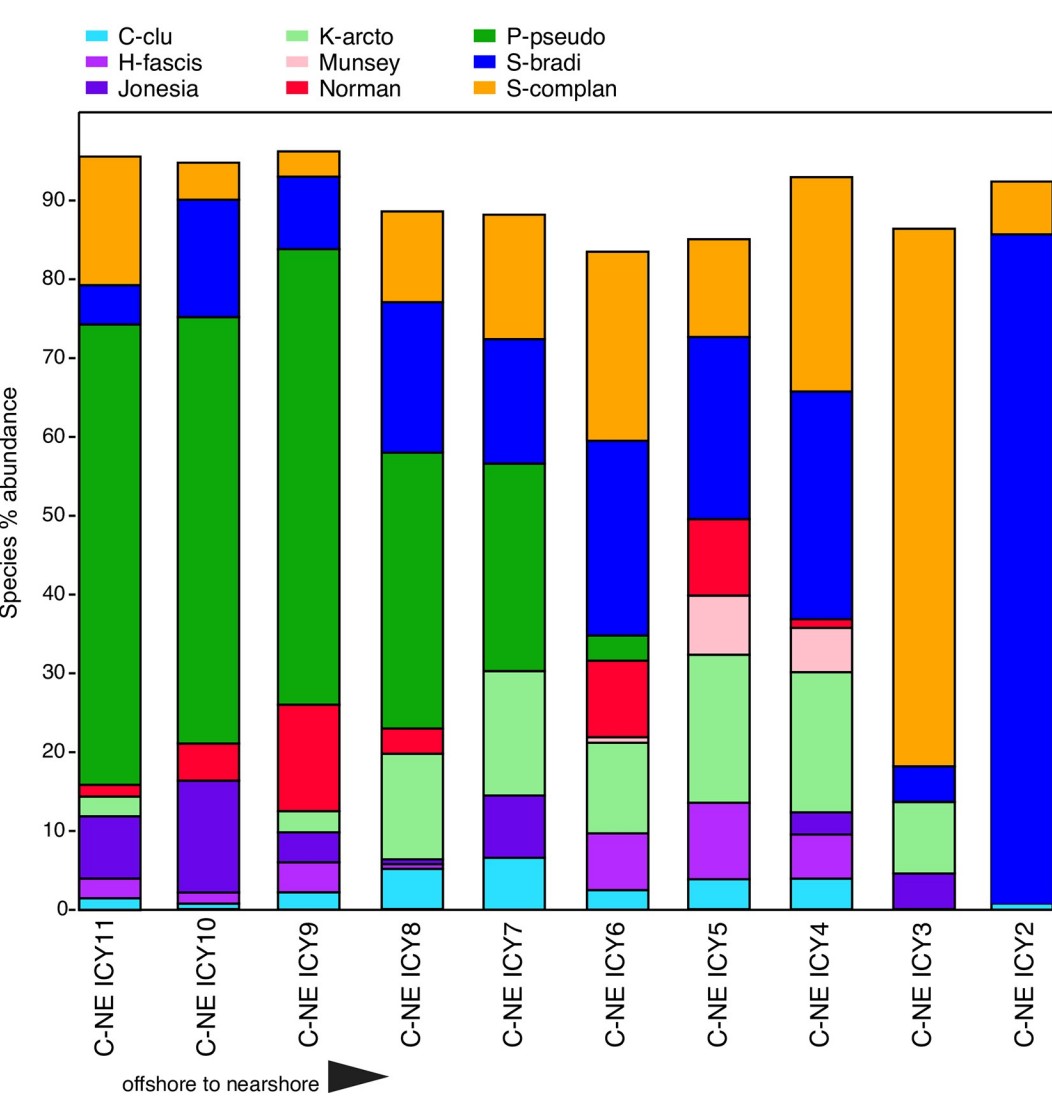

**Fig 10. Primary species, Northeast Chukchi Sea—Icy Cape.** Stack plot of the 9 most abundant species comprising ~90% of the ostracode assemblage at the 2018 Icy Cape (ICY) sampling transect (sites on Fig 2 map), northeast Chukchi Sea, (n = 10 samples with ≥75 specimens/sample [except ICY2 that contained 22 specimens], 1,554 total specimens). Species not included on plot that each represent >1–2% of transect assemblage total include *A. dunelmensis*, *Argilloecia* sp., and *E. concinna*.

DBO2-Chirikov Basin (Fig 7A), Ledyard Bay (Fig 9), Icy Cape (Fig 10) and DBO5-Barrow Canyon (Fig 12) transects, ostracode biofacies do not show a continuous composition but instead a faunal transition from nearshore to offshore stations. The ecoregion plots show distinct abundance changes in indicator species, which were sometimes acutely distinct, based on spatial environmental changes in water mass, sediment properties and food type. These descriptions of ostracode faunal assemblages can serve as a baseline to examine future meiofaunal changes in the BCB Seas.

**Chirikov Basin—DBO2.** The DBO2 transect in the Chirikov Basin (Fig 7A) is an example of ostracode indicator species clearly divided by water masses of differing productivity.

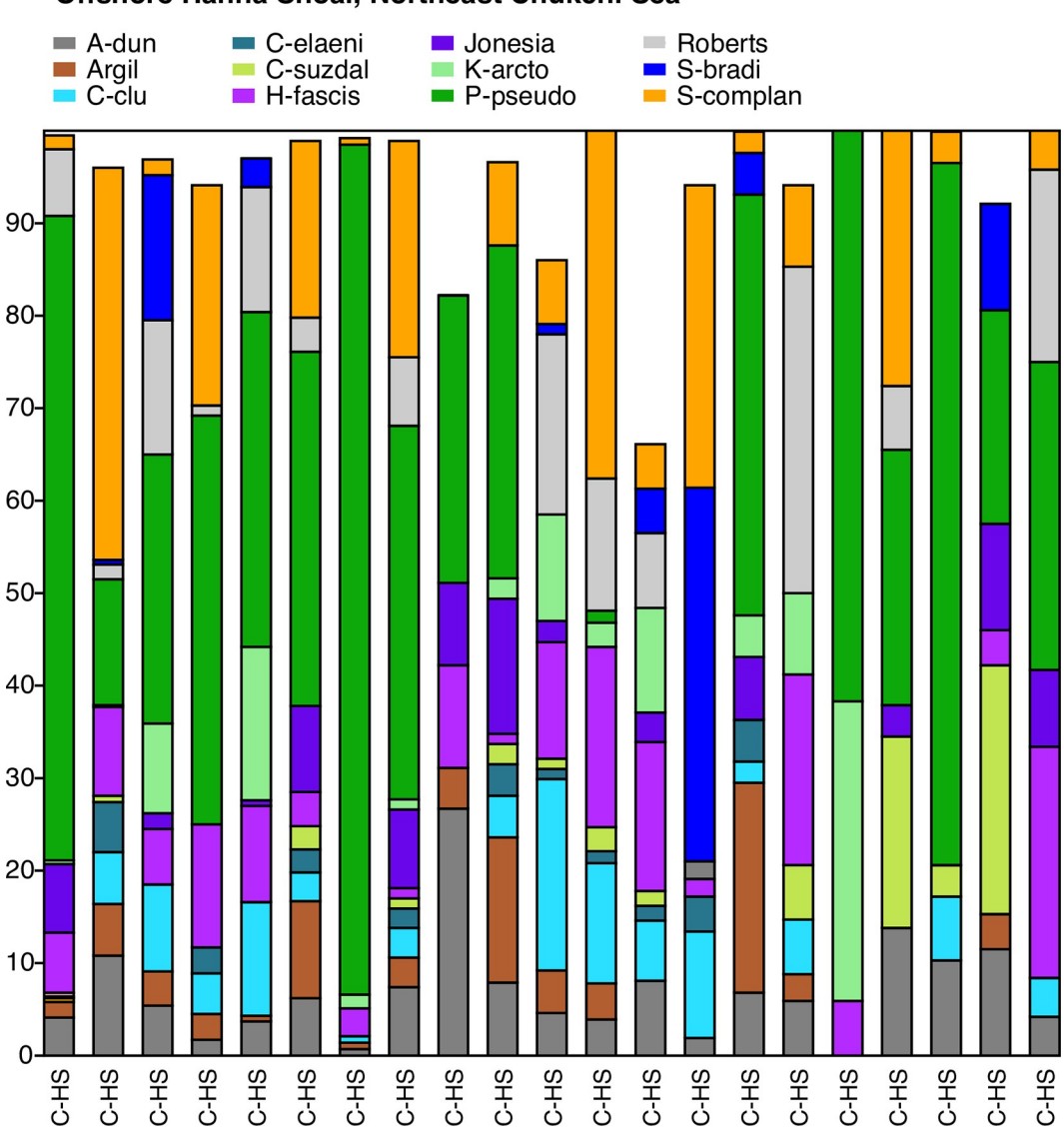

**Fig 11. Primary species, Northeast Chukchi Sea–Hanna Shoal.** Stack plot of the 11 most abundant species and one genus (*Argilloecia* spp.) comprising ~96% of the ostracode assemblage in the Hanna Shoal region offshore in the northeast Chukchi Sea (sites on Fig 2 map) from years 2012 to 2013, (n = 21 surface sediment samples with ≥30 specimens/sample, 2,647 total specimens in an average water depth of 51m).

Stations DBO2-1 and 2–4 are the western-most locations, and are in the path of AW (Fig 2). Major differences between the water masses include higher average values of sediment chloro-phyll-a and salinity (Fig 7B) to the west because the more eastern samples are underlain by BSW or a combination of ACW above and BSW below. Integrated primary production esti-mates in the Chirikov Basin range from ~80 g C m$^{-2}$ yr$^{-1}$ on the interior shelf to up to 480 g C m$^{-2}$ yr$^{-1}$ in AW [11]. *N. leioderma* (red) is more prevalent in AW. The eastern side of the Basin (stations DBO2-2 and 2–5) is dominated by *S. bradii* (blue) and *E. concinna* (brown), with increasing numbers of *S. ikeyai* (light blue) in 2018 samples that may reflect slightly higher average bottom water temperatures (nearly 2˚C, Fig 4B). The abrupt change in faunas from

**Barrow Canyon / DBO5  Northeast Chukchi Sea**

**Fig 12. Primary species, DBO5 –Barrow Canyon.** Stack plot of the 11 most abundant species and one genus (*Finmarchinella*), comprising ~80% of the ostracode assemblage, in DBO5-Barrow Canyon region (sites on Fig 2 map) from years ranging from 2011 to 2018 (n = 31 surface sediment samples with ≥30 specimens/sample, 2,951 total specimens). Other faunas at stations 5–9 and 5–10 that account for the species not shown on the stack plot include *A. dunelmensis*, *J. acuminata* and *H. fascis*, and *R. tuberculatus*.

east to west related to water mass channels is very consistent on interannual timescales at these sampling locations, with the exception of DBO2-4. Earlier samples from years 2001 and 2003 do not have *N. leioderma* as the dominant fauna at DBO2-4, and samples from years 2016 and 2018 do. We speculate that *N. leioderma* may have expanded its population perhaps due to increasing carbon export to the benthos and/or changes in current flow that affected sediment grain size. Since satellite monitoring began in 1979, the length of the ice-free season in the Chirikov Basin has increased by ∼25 days, stimulating greater primary productivity due to a longer growing season [74, 75]. With *N. leioderma* as the established dominant fauna in the northern Bering Sea around the St. Lawrence Island polynya [31], additional light and productivity may be factors that contribute to its expansion. Another major feature of this area is strong currents that create coarse seafloor sediments consisting of primarily (>75%) 0–4 phi-sized sand, which *N. leioderma* favors. Macrofaunal biomass associated with finer-grained

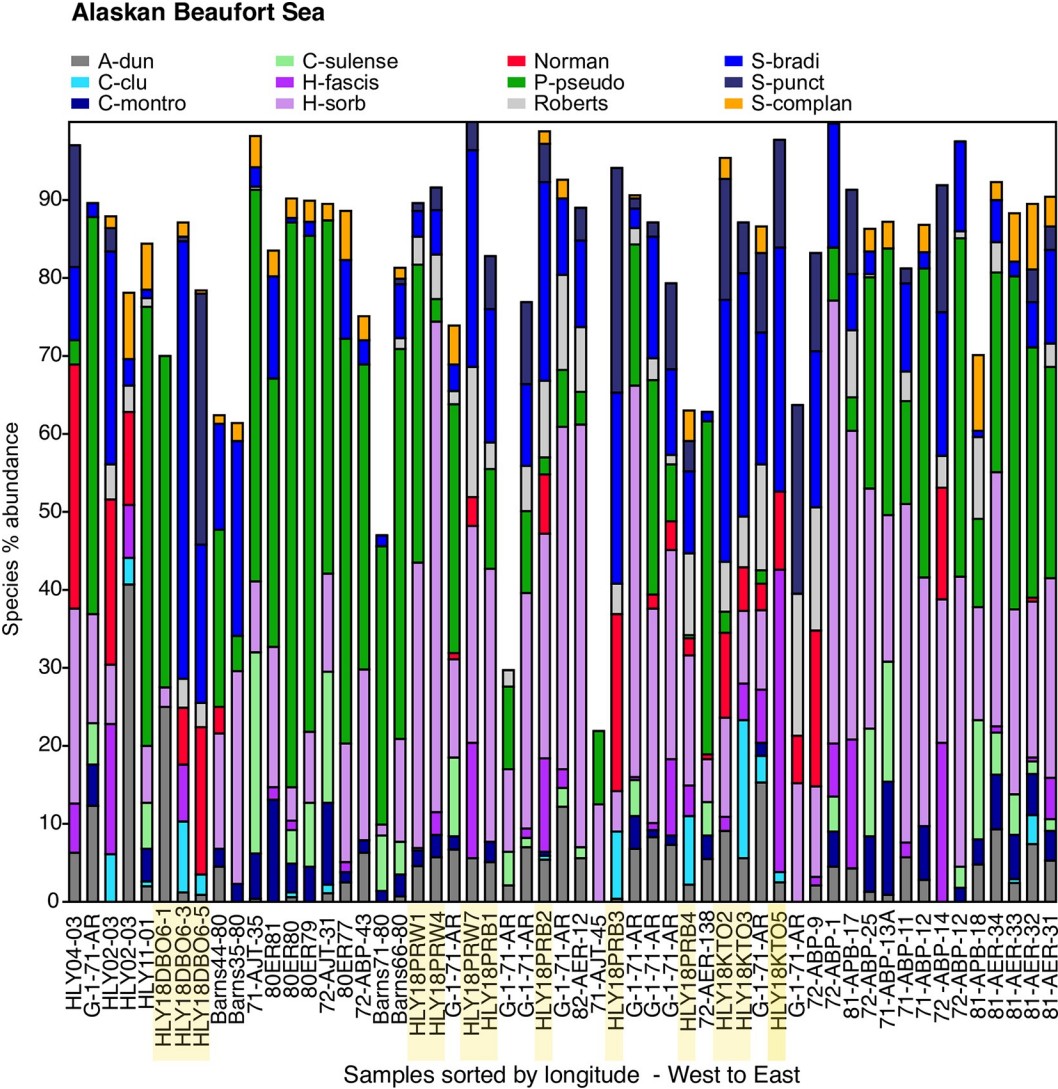

**Fig 13. Primary species, Alaskan Beaufort Sea.** Stack plot of the 12 most abundant species comprising ~85% of the ostracode assemblage in a narrow band of the Alaskan Beaufort Sea (sites on Fig 4 map) within 100 km of shore at 20-90m water depth from years 1971 to 2018, (n = 55 surface sediment samples with ≥30 specimens/sample, 6,617 total specimens). Samples collected in 2018 are highlighted in yellow. Other species not shown on the stack plot that account for 1–2% of the assemblage include *Rabilimis septentrionalis*, *Pteroloxa chaunensis*, *E. concinna*, and several *Cytheropteron* species. Important species in the Bering and Chukchi that exhibited either very low abundance (<1%) or are absent in the Beaufort Sea at 20-90m water depth are: *S. ikeyai.*, *M. mananensis*, *Finmarchinella spp.*, *K. arctoborealis*, *H. clathrata*, *H. emarginata*, *J. acuminata*, *S. affinis*, *S. mainensis*, and *S. undata*.

sediments has significantly declined at sampling station DBO2.4 from 1999–2015 [16]. At this site, *N. leioderma* comprised >50% of the assemblage composition in 2018 and 2016 compared to <10% in 2001 and 2004 (Fig 7A).

**Southeast Chukchi Sea—DBO3.** At the nearest-to-shore stations, DBO3-1 to 3–4, the ostracode biofacies during 2014 to 2018 have been extremely consistent, not only in species present but species proportions as well (Fig 8). Species at DBO3-1 to 3–4 stations are in the path of ACW (temperatures 6˚–10˚C) that reach to the seafloor [76]. Higher sediment grain sizes at these near shore stations are consistent with the presence of the ACC. These nearshore

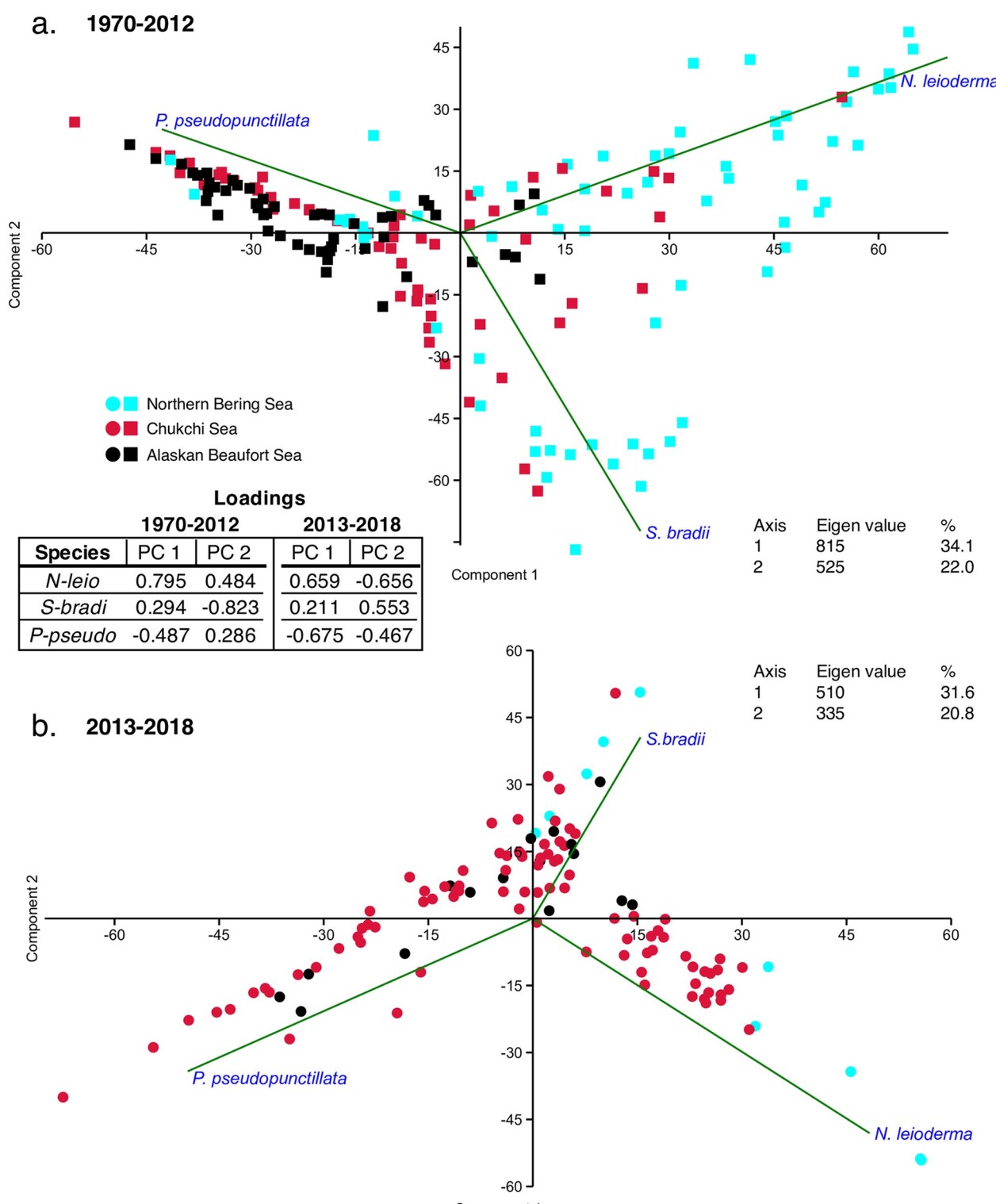

**Fig 14. Principal correspondence analysis of ostracode assemblages by year groups.** Synoptic PCAs using the biogeographic dataset (samples located on Fig 1 map) from the BCB grouped by the three regions and two collection periods, a.) 1970–2012 and b.) 2013–2018.

stations experience great variability in seasonal bottom water temperatures (~10˚C) and are dominated by *N. leioderma* (35%) with secondary taxa *S. bradii* (14%), *S. ikeyai* (13%), *E. concinna* (6%), *M. mananensis* (3%) and *Finmarchinella* spp. (5%). *Paracyprideis pseudopunctillata* is also present in many sample years in low abundance (2%). *H. sorbyana* (3%), an extremely euryhaline species [66, 77], appears only at nearest-to-shore DBO3-1 sites, and reflects the fresher water (average 31 salinity) of ACW. DBO3-1 to 3–4 are hotspots of ostracode abundance (samples averaged 235 specimens/sample) compared to sites further south or north. Despite high current velocities at these near shore stations, as indicated by coarser sediment grain sizes (75% phi 0–4 [±9]), sediment chlorophyll-a averaged 18 mg/m$^2$ (±11) at DBO3-1 to DBO3-5 sites from years 2014 to 2018, which supported consistently high abundances of *N. leioderma* and other ostracode species.

Samples just north of the Bering Strait (denoted as "N of St" on Fig 12) and DBO3-5 are offshore in a different summer water mass than DBO3 sites closer to shore. This water mass change is indicated by a different ostracode biofacies with fewer species overall and high proportions of *N. leioderma* and *Semicytherura* spp., particularly *S. complanata* and *S. mainensis*.

Ostracodes are rare or absent in sites further offshore (DBO3-6 to DBO3-8) in the path of BSW as sediment composition shifts to finer-grained sediments (slower current speeds, ~50–93% ≥5 phi) and higher TOC (>1%) values. This pattern is consistent over multiple years of sampling. Benthic macrofaunal biomass is very high at these offshore sites, with abundant production settling to the sediments [51].

**Central Chukchi Sea—Ledyard Bay 2018.** While the ostracode biofacies at most sampling sites along the 2018 Ledyard Bay transect are similar to those at DBO3 sites (except *S. ikeyai* is lacking at Ledyard Bay [2 individuals]), the Central Channel bearing BSW north to the shelf break (Fig 2) cuts across this sampling transect and creates sediment grain size, production and temperature gradients between the sample locations that drastically change ostracode assemblage composition. The station LB-5, in the path of ACW proximal to shore, has coarse sediments (75% 0–4 phi), a summer bottom water temperature at the time of sampling of 8.7˚C, and 31 salinity. Subarctic species *N. leioderma* (20%), *M. mananensis* (6%) and *Semicytherura* species (32%) occupied this site (Fig 9). At the next site, LB-7, sediments change to finer silt (59% ≥5 phi) and *P. pseudopunctillata* becomes dominant with *Palmenella limicola* and *S. complanta*. *N. leioderma* is absent. Further offshore there is a grain size shift back to coarser sediments east of Herald Shoal (62% in the 0–4 phi category at LB-13, and of that, 25% in the 0–2 phi category) where *N. leioderma*, *S. bradii*, *M. mananensis* and *Kotoracythere arctoborealis* become predominant. In addition, the highest primary productivity rates in the Chukchi Sea are consistently observed in the central part of the shelf along the dividing line between the warm Alaskan Coastal Current to the east and in the colder nutrient-rich Bering shelf waters to the west [78]. *N. leioderma's* increasing abundance at LB-11 and LB-13 may reflect a fresh, more ample food source. TOC increases toward the offshore stations and sediment chlorophyll-a, although seasonally and yearly highly variable is highest at LB-13 (20 mg/m$^2$) as is *N. leioderma's* (30%) abundance. Along the transect, temperature declined with distance from the coast from 8.7˚C at LB-5 to 2.4˚C at LB-13, where *N. leioderma*, *S. bradii*, *S. complanata*, *M. mananensis* and *K. arctoborealis* persist. This transect may be an example of how subtle changes in bathymetry and current flow can result in changes in food availability and sediment texture in very localized areas that affect the spatial distribution of benthic communities [73] and, likewise, ostracode biofacies.

**Northeast Chukchi Sea—Icy Cape 2018.** The 2018 Icy Cape transect shows the influence of bottom water mass and sediment texture on assemblage composition. Bottom water temperature closest to shore at ICY2 was 1.4˚C, but more offshore stations were colder with temperatures of 0˚ to -1˚C (Fig 10). In the northeast Chukchi Sea, sea-ice coverage usually extends

Selected species time series, Chukchi Sea, 2000-2018

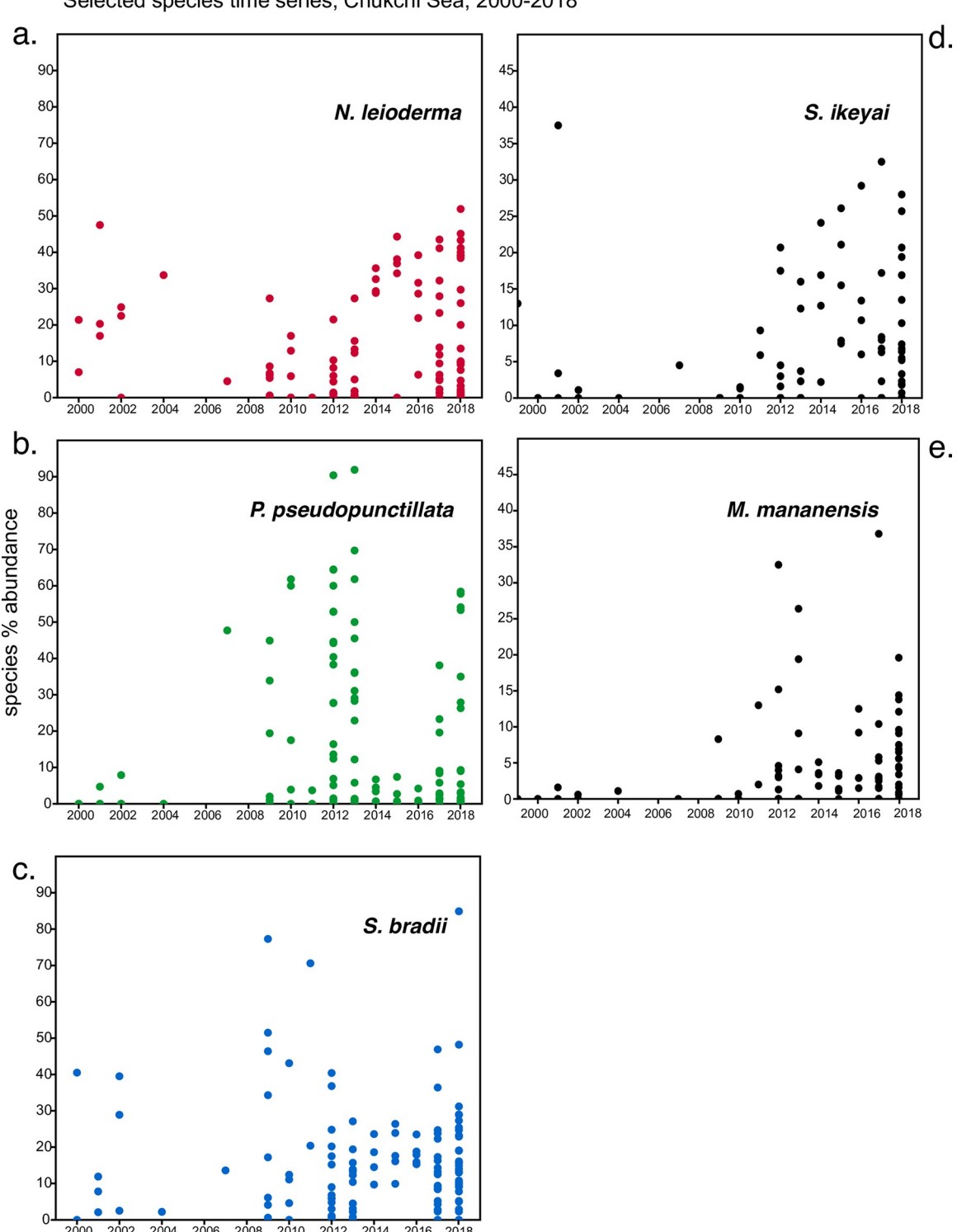

**Fig 15. Time series of key species in Chukchi Sea.** a.) *N. leioderma*, b.) *P. pseudopunctillata*, c.) *S. bradii*, d.) *S. ikeyai*, e.) *M. mananensis*.

into the summer months (July- August), with areas of localized persistence possible throughout the summer. Winter water was still present on the seafloor during late summer sampling. Close to shore, grain size at ICY2 to ICY5 was coarse (45% to 76% in phi 0–4 category), where *S. bradii* and *S. complanata* were predominant and *P. pseudopunctillata* was absent. The presence of *P. pseudopunctillata* beginning at ICY6 and steadily increasing to dominate (25% to 60%) the assemblages at ICY9, ICY10 and ICY11 is indicative of a seafloor sediment transition to very fine grains, ranging from 63% to 87% ≥5 phi. In addition to suitable habitat, persistent cold temperature supports *P. pseudopunctillata*. *Jonesia acuminata* is correlated to near-freezing temperatures and fine sediments [31], and this species has a small but notable presence at the ICY7 to ICY11 stations.

**Offshore Hanna Shoal, Northeast Chukchi Sea.** Heterogeneous seafloor contours direct BSW with high nutrient concentrations to the Hanna Shoal area [79]. It is a hotspot for benthic productivity that supports marine mammals [15]. The ostracode biofacies includes *P. pseudopunctillata* (37% of assemblage) with secondary species *S. complanata* (14%), *H. fascis* (8%), *R. tuberculatus* (7%) and *C. cluthae* (5%; Fig 11). It is very similar to the assemblage of offshore Icy Cape sites except this biofacies includes greater proportions of *A. dunelmensis* (6%) and *R. tuberculatus*, and *Argilloecia* sp. (4%), which are species associated with finer-grained sediments (Fig 6B). *Normanicythere leioderma* is rare (1%). *Paracyprideis pseudopunctillata* and *H. fascis* are euryhaline and can tolerate fluctuating salinities. At the time of summer collection, the average salinity at sample locations was 33, bottom temperature, -1.7˚C and water depth, 51m. This area lacks any strong gradient from currents, as sea ice tends to be trapped over Hanna Shoal, so dense bottom-confined pools formed by winter water tend to stagnate [38, 39, 80]). Sustained cold temperatures, fine-grained sediments, and relatively higher salinities describe this frigid-water assemblage.

**Barrow Canyon/DBO5.** Sampling stations on the DBO5 transect have significant changes in depth, from ~50m at DBO 5–1 and DBO5-2, ~90m at DBO5-3 and deepening further to ~130m at DBO5-5, ~120m at DBO5-6, ~90m at DBO5-7 and ascending to 60-70m depths at DBO5-8 to DBO5-10 (Fig 2). Although seasonally variable, the deeper locations had much higher sediment chlorophyll-a (18–26 mean mg/m$^2$) values measured in summer, deposited by variable strong currents flowing through the Canyon. *N. leioderma* increases in abundance at these sites, particularly DBO5-6 and 5–7 stations, which historically also have high TOC values (~2%; Fig 12). These offshore stations are usually characterized by bottom water temperatures <0˚C, however nearshore DBO5-1 and 5–2 sites have experienced summer warming in recent years with temperatures reaching 4–7˚C (Fig 4D). Station DBO5-1 is influenced by ACW, which can reach the seafloor. These nearshore stations are dominated by *S. bradii* and cold-temperate/subarctic taxa *M. mananensis* and *S. ikeyai*, which decline in abundance moving seaward towards DBO5-5. (Note: Sites DBO5-4 and 5–8 are not represented due to low ostracode abundance of <30 specimens/sample.) At stations DBO5-6 to 5–10, seafloor sediments are extremely fine-grained silt, typically ≥90% 5phi. These sediments are the finest-grained of any ecoregion, and they are occupied by ostracode faunas that require this habitat along with near-freezing temperatures, notably *P. pseudopunctillata*, *S. complanata* and *C. cluthae*. At station DBO5-9 there is a distinct faunal transition to these species. Cold (<0˚C), dense (~33–34 salinity) WW can remain at these stations during summer [81]. Occasionally, warm (>-1.2˚C), salty (>33.6) Atlantic water is upwelled into the Canyon and fills the deepest part of the area [82].

**Alaskan Beaufort Sea.** The Alaskan Beaufort continental shelf is a highly variable environment, influenced by seasonal discharge from the Colville and Mackenzie rivers that supply fresh water and deposit terrigenous material but lacks strong gradients within the examined narrow area of 20-90m water depth (Fig 3; sample locations were limited to within 100 km of

shore, and included intermittent years between 1971 and 2018). Bottom water salinity at the sampling sites during the time of collection averaged 31. In addition to river inflow, the Beaufort shelf also receives dense "winter-transformed" Pacific water [83, 84] that drains off the Chukchi shelf. This water enters the Beaufort Sea as a shelf-break current that turns eastward from Barrow (Fig 2); [85]. As an area that experiences dramatic seasonal fluctuations, particularly in salinity, two species are overwhelmingly suited to these conditions: *P. pseudopunctillata* and *H. sorbyana*, cryophilic taxa correlated with euryhaline conditions (Fig 13); [19, 20]. These two species are reported by Stepanova et al. [19, 20] as adapted to the inner shelf, river-affected zone of the Kara and Laptev seas where salinities range between 26 and 32. The eastern Alaskan Beaufort is influenced by the Mackenzie River, resulting in finer sediment grain sizes and better sorting in that region [86, 87], which correlates to greater abundance of *P. pseudopunctillata*. For 2018 PRW and PRB transect samples (sites on Fig 3 map) that are most influenced by Colville River inflow, *H. sorbyana* is more prevalent than *P. pseudopunctillata*. Secondary species in the Beaufort Sea assemblage include *S. bradii*, *S. punctillata*, *R. tuberculatus*, *H. fascis*, *and A. dunelmensis*. *S. complanata* is present in low (~2%) abundance in these Beaufort Sea assemblages.

Despite limited temporal data in this area, *N. leioderma* and *S. bradii* are more prevalent in 2018 samples than in samples collected in comparable locations during the 1970s and 1980s where *P. pseudopunctillata* and *H. sorbyana* comprised the major proportion of the assemblage (S3 Fig). This may be due to recent declines in ice persistence and increases in nearshore temperatures or changes in sediment composition that may be incompatible with *P. pseudopunctillata's* preferences for near-freezing temperatures and finer grained sediments. However, sediment transport by ice rafting can be significant on the shelf, and can carry gravel and sand particles from the coast onto the shelf [87]. There are insufficient data available to determine if a change in sediment grain size is a factor. The Beaufort shelf, while not as productive as the Chukchi Sea, can have locally high primary production rates associated with the ice edge, reaching 200mg C m$^{-2}$ d$^{-1}$ [11]. The retreating sea-ice edge, in addition to upwelling, provides nutrients for phytoplankton growth. The number and strength of upwelling events in the Alaskan Beaufort Sea has increased over the past 25 years [88, 89]. Greater fresh phytoplankton production may be an explanation for the increasing appearance of *N. leioderma* in 2018 samples (S3 Fig), but our new data must be considered cautiously given the single year of samples.

## Temporal trends in BCB Sea indicator species

Criteria used to assess whether environmental factors are driving changes in benthic communities include the arrival of taxa characteristic of a temperate climate zone or shifting dominance or abundance of taxa [90]. Species can extend north and south (and east to west) to the point where certain maximum or minimum tolerable temperatures are reached [23].

Due to temporal and spatial gaps in sampling, we could confidently evaluate temporal changes only in the dominant species, *N. leioderma*, *S. bradii* and *P. pseudopunctillata*. Two PCA comparisons by years (1970–2012 and 2013–2018) showed that the dominant species have remained fairly consistent in the BCB Seas (Fig 14). Interannual abundances of the dominant species in the Chukchi Sea from 2000–2018 did not reveal statistically significant changes, but two indicator species, *S. ikeyai* and *M. mananensis*, show incipient increases (Fig 15).

Dispersal of meiofauna into new areas likely lags behind sediment and water column changes. Benthic marine ostracodes are transported by ocean currents [91]. Considering that the movement of Pacific water from the Aleutian Passes to Bering Strait takes more than a year [5] and an additional ~4 months to transit to Barrow Canyon in the Chukchi Sea [40], benthic meiofaunal migration from the north Pacific to the Arctic could take a number of years. Over the shallow shelves of the northern Bering and Chukchi Seas, seasonal ice coverage cools the

entire water column to temperatures below 0˚C. These cold temperatures limit the northern distribution of subarctic populations of groundfish [92] and may also serve as a barrier inhibiting the mixing of Arctic and N. Pacific faunas through the Bering Strait [71]. It is more likely that cold temperate species already inhabiting restricted inner bay areas like Norton Sound and coastal polynyas along Alaska will increasingly be able to extend their range and abundance as conditions become more suitable for them under changing climate scenarios.

## Conclusions

We examined benthic ostracode assemblages collected during research cruises from 1970–2018 in the northern Bering, Chukchi and Alaskan Beaufort Seas. We focused on identifying indicator species and their associated ecological preferences. In particular, ostracode abundance and distribution were related to bottom temperature, salinity, organic carbon deposition (sedchla, TOC, C/N), and sediment substrate.

The variability in ostracode assemblage composition throughout the BCB Seas was linked qualitatively to localized (summer seasonal) environmental patterns. We found south-to-north (Chirikov Basin to Northern Chukchi to Beaufort) changes in dominant ostracode species and assemblage composition. Over the large biogeographic scale, four ostracode biofacies were identified in the study area primarily tied to the transit of summer water masses. Distinct changes of ostracode biofacies in nearshore to offshore transects were related to water mass properties likely in combination with food sources and sediment substrate. This study is among the first to highlight changes in ostracode abundance and species over small spatial scales in response to changes in water mass properties and productivity.

Six indicator species were identified by correspondence and multivariate analyses based upon their abundance and correlations with distinct ecological parameters by which they are defined:

*Normanicythere leioderma*–opportunistic subarctic species correlated with ample, high-quality production (exported phytoplankton) as food source; sandy to pebbly to gravelly sediments;

*Paracyprideis pseudopunctillata*–euryhaline cryophilic species associated with WW presence and sea ice, fine grained sediments, sustained frigid temperatures (≤1˚C), TOC with high phytodetritus;

*Sarsicytheridea bradii*—habitat generalist, eurytopic species that capitalizes in highly dynamic environments, higher frequency in sandy sediments;

*Semicytherura complanata*–frigid temperatures (≤1˚C), normal marine salinity, no discernable sediment preference;

*Schizocythere ikeyai*–cold-temperate species, warmer temperatures, ACW, sandy sediments;

*Munseyella mananensis*–cold-temperate species, warmer temperatures, ACW, sandy sediments.

This study found a consistent link between areas of high sediment chlorophyll, coarse sediment grain size and the abundance of *N. leioderma*. Despite wide tolerances, it may be a sensitive indicator of new production reaching the benthos and/or enhanced food supply. In future paleoceanographic studies using fossil ostracode faunal assemblage data, *N. leioderma* may signal the influence of nutrient-rich Pacific water flowing in through the Bering Strait and a food supply consisting of newly settled phytoplankton. *Paracyprideis pseudopunctillata* is positively

correlated to very fine-grained sediment textures with high TOC values of refractory (detrital) organic carbon food sources.

During the last decade, we document an incipient increase in two secondary cold-temperate species, *S. ikeyai* and *M. mananensis*, that may be increasing. Preliminary interpretation of this finding may reflect recent increases in coastal and mid-shelf bottom water temperatures and/or carbon flux to the benthos.

Continued monitoring of temperature-sensitive ostracode species in the BCB Seas is necessary to provide information on annual and decadal variability in species distributions. This analysis of modern species ecology can help interpretation of ostracode faunal data from sediment cores in relation to past and future ocean changes. This study contributes new species ecology and further validates the sensitivity and application of ostracode fauna as biomonitors and proxies for specific environmental conditions. These results provide a baseline for assessing the effects of future water mass changes and productivity on benthic ostracode communities in the Pacific Arctic.

## Supporting information

**S1 Fig.** a. Faunal abundance of selected taxa in relation to near-bottom temperature during summer sediment collection in the northern Bering and Chukchi Seas (n = 211, 26,170 total specimens, 1990–2018, ≥30 specimens/sample). b. Faunal abundance in relation to salinity. c. Faunal abundance in relation to sediment type. Ostracode species abundance plotted against the percent sediment modal grain size of phi 0–4, where 0 represents gravel and rocks, 1 = coarse sand, 2 = medium sand, 3–4 = finer sand. Phi ≥5 (not shown) represents the very fine silty mud and clay sediment fraction typical of offshore or interior areas of the continental shelf.
(PDF)

**S2 Fig.** Faunal abundance plotted against three different sources of sediment carbon, which may suggest preferred food sources: a. sediment chlorophyll-a (sedchla), b. total organic carbon (TOC) and c. carbon to nitrogen (C/N) ratios.
(PDF)

**S3 Fig. Principal Correspondence Analysis (PCA) of Alaskan Beaufort Sea ostracode assemblages.** Sample sites are designated by collection years (legend symbols) and major taxa (green lines with species names labeled in blue). The dominant species in the Alaskan Beaufort Sea are *H. sorbyana*, *P. pseudopunctillata*, *S. bradii* and, in samples collected in 2018, *N. leioderma*.
(PDF)

**S1 Table. Subset of data from the Arctic Ostracode Database (AOD-2020; Cronin et al., 2021), supplemented with sample sediment parameters, used in this study.**
(XLSX)

**S2 Table. Six ecologically defined species in the northern Bering, Chukchi, and Beaufort Seas based on multivariate correlations with environmental factors (\* = from listed ecological references).** Scanning electron microscope (SEM) photos of species are taken from Gemery et al., 2015.
(PDF)

## Acknowledgments

We thank A. Ruefer, N. Vaka and S. Watson for assistance with sample processing and E. Fachon for collection of 2018 Beaufort Sea samples during the HLY18-03 expedition. We also thank the field technicians in the Grebmeier/Cooper laboratory at CBL/UMCES for water column and sediment collections from multiple time-series cruises used in this study. We would also like to thank J. Keith, M. Robinson, G. L. Wingard, and two anonymous reviewers for critical comments that improved earlier versions of the manuscript. Any use of trade, firm, or product names is for descriptive purposes only and does not imply endorsement by the U.S. Government.

## Author Contributions

**Conceptualization:** Laura Gemery.

**Data curation:** Laura Gemery, Lee W. Cooper.

**Formal analysis:** Laura Gemery, Harry J. Dowsett.

**Funding acquisition:** Laura Gemery, Thomas M. Cronin, Lee W. Cooper, Jacqueline M. Grebmeier.

**Investigation:** Laura Gemery, Lee W. Cooper, Harry J. Dowsett, Jacqueline M. Grebmeier.

**Methodology:** Laura Gemery.

**Project administration:** Thomas M. Cronin.

**Supervision:** Thomas M. Cronin, Lee W. Cooper.

**Validation:** Laura Gemery.

**Visualization:** Laura Gemery.

**Writing – original draft:** Laura Gemery.

**Writing – review & editing:** Laura Gemery, Thomas M. Cronin, Lee W. Cooper, Harry J. Dowsett, Jacqueline M. Grebmeier.

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
