## [Decision Letter · Decision Letter 0]

6 Apr 2021

PONE-D-21-04622

Biogeography and ecology of Ostracoda in the U.S. northern Bering, Chukchi, and Beaufort Seas

PLOS ONE

Dear Dr. Gemery,

Thank you for submitting your manuscript to PLOS ONE. After careful consideration, we feel that it has merit but does not fully meet PLOS ONE’s publication criteria as it currently stands. Therefore, we invite you to submit a revised version of the manuscript that addresses the points raised during the review process.

I have now received the comments of two external reviewers; both of them are really positive on the novelty and wider perspective of the present Ms. They put forward only a few suggestions on the statistical part and possibly adding plate(s).

We look forward to receiving your revised manuscript.

Kind regards,

Fabrizio Frontalini

Academic Editor

PLOS ONE

Journal Requirements:

2. Please include your tables as part of your main manuscript and remove the individual files. Please note that supplementary tables should remain uploaded as separate "supporting information" files.

"Financial support for sample collections was provided by grants to JMG and LWC from the NSF Arctic

Observing Network program (1204082, 1702456 and 1917469), NOAA Arctic Research Program

(CINAR 22309.07 and 25984.02, https://arctic.noaa.gov/), PacMARS (Pacific Marine Arctic

Regional Synthesis) project funded by Shell Exploration and Production and ConocoPhillips, and

administered and managed by the North Pacific Marine Research Institute (NPMRI Arctic Project

A01) through the North Pacific Research Board, and with oversight from the National Science

Foundation (NSF) Division of Polar Programs, and BOEM funding through the COMIDA Hanna

Shoal Project (UTA11-000872); and the USGS Climate & Land Use R&D Program."

Additionally, because some of your funding information pertains to commercial funding, we ask you to provide an updated Competing Interests statement, declaring all sources of commercial funding.

In your Competing Interests statement, please confirm that your commercial funding does not alter your adherence to PLOS ONE Editorial policies and criteria by including the following statement: "This does not alter our adherence to PLOS ONE policies on sharing data and materials.” as detailed online in our guide for authors  http://journals.plos.org/plosone/s/competing-interests.  If this statement is not true and your adherence to PLOS policies on sharing data and materials is altered, please explain how.

Please include the updated Competing Interests Statement and Funding Statement in your cover letter. We will change the online submission form on your behalf.

4. We note that Figures 1-3 in your submission contain map/satellite images which may be copyrighted. All PLOS content is published under the Creative Commons Attribution License (CC BY 4.0), which means that the manuscript, images, and Supporting Information files will be freely available online, and any third party is permitted to access, download, copy, distribute, and use these materials in any way, even commercially, with proper attribution. For these reasons, we cannot publish previously copyrighted maps or satellite images created using proprietary data, such as Google software (Google Maps, Street View, and Earth). For more information, see our copyright guidelines: http://journals.plos.org/plosone/s/licenses-and-copyright.

4.1.    You may seek permission from the original copyright holder of Figures 1-3 to publish the content specifically under the CC BY 4.0 license. 

4.2.    If you are unable to obtain permission from the original copyright holder to publish these figures under the CC BY 4.0 license or if the copyright holder’s requirements are incompatible with the CC BY 4.0 license, please either i) remove the figure or ii) supply a replacement figure that complies with the CC BY 4.0 license. Please check copyright information on all replacement figures and update the figure caption with source information. If applicable, please specify in the figure caption text when a figure is similar but not identical to the original image and is therefore for illustrative purposes only.

Reviewers' comments:

Reviewer's Responses to Questions

**Comments to the Author**

1. Is the manuscript technically sound, and do the data support the conclusions?

Reviewer #1: Yes

Reviewer #2: Yes

2. Has the statistical analysis been performed appropriately and rigorously? 

Reviewer #1: Yes

Reviewer #2: Yes

3. Have the authors made all data underlying the findings in their manuscript fully available?

Reviewer #1: Yes

Reviewer #2: Yes

4. Is the manuscript presented in an intelligible fashion and written in standard English?

Reviewer #1: Yes

Reviewer #2: Yes

5. Review Comments to the Author

Reviewer #1: In my opinion, the paper is very interesting in approach and it furnishes relevant ostracod data from the Pacific-Arctic region, a key area of investigation also in the view of the climate change.

I’ve found especially interesting (i) the multivariate analyse of a big dataset composed of ca. 300 samples, collected during the last decades from continental shelves characterized by different physical-chemical parameters and (ii) the potentiality to use the obtained results to infer more robust paleoenvironmental reconstructions and to track potential changes in ecosystems under changing climate conditions.

Data interpretations and discussion are well justified and represent a logical product of a comprehensive and multi-faceted work.

I list below few comments, that I hope might be helpful to the Authors.

• I suggest to move the aims of the paper from lines 100-108 to the end of Introduction

• Line 173: “≥5 phi = fine silts” maybe “fine silts and clay”?

• Was the ostracod matrix treated before the application of mltivariate analyses?

• Concerning DCA (Fig. 6a), I wonder how the Authors traced the limits of the areas/groups and why a group of samples remained outside these groups, at the left edge of the biplot. I think that these points should be explained in text.

• I suggest to integrate the section “Time-series analysis” into the discussion section “Temporal trends in BCB Sea indicator species”.

• Why any environmental gradient has been identified along the transect “Offhore Hanna Shoal” and “Alaskan Beaufort Sea”? I think that these points deserve a little bit of discussion.

• Figure 1: I suggest to insert a map for an easier localization of the studied area.

• I suggest to add the ostracod matrix used for multivariate analyses as supplementary material.

Yours faithfully

Reviewer #2: Dear Gemery and co-authors, your manuscript "Biogeography and ecology of Ostracoda in the U.S. northern Bering, Chukchi, and Beaufort Seas" is very well written and organised. I have annotated a pdf with some comments mainly about statistical analysis. I suggest to add a plate with the indicator taxa and to revise the reference list.

6. PLOS authors have the option to publish the peer review history of their article (what does this mean?). If published, this will include your full peer review and any attached files.

Reviewer #1: No

Reviewer #2: No

---

## [Author Response · Author response to Decision Letter 0]

16 Apr 2021

5. Review Comments to the Author

Reviewer #1: 

In my opinion, the paper is very interesting in approach and it furnishes relevant ostracod data from the Pacific-Arctic region, a key area of investigation also in the view of the climate change.

I’ve found especially interesting (i) the multivariate analyse of a big dataset composed of ca. 300 samples, collected during the last decades from continental shelves characterized by different physical-chemical parameters and (ii) the potentiality to use the obtained results to infer more robust paleoenvironmental reconstructions and to track potential changes in ecosystems under changing climate conditions.

Data interpretations and discussion are well justified and represent a logical product of a comprehensive and multi-faceted work.

I list below few comments, that I hope might be helpful to the Authors.

• I suggest to move the aims of the paper from lines 100-108 to the end of Introduction

Author’s response: We think this is a good suggestion, and have made this change

• Line 173: “≥5 phi = fine silts” maybe “fine silts and clay”?

Author’s response: Yes, we have added “and clay” here

• Was the ostracod matrix treated before the application of multivariate analyses?

Author’s response: No, there was no special treatment; The species counts were converted into species percent of the total assemblage and then used directly. The Bering-Chukchi-Beaufort dataset is a subset from the Arctic Ostracode Database-2020 (Cronin et al., 2021). As explained in the Methods section, we used only samples that contained ≥30 specimens, as that was the cutoff we deemed large enough to contain a representative sample of ostracodes living at a particular location.

• Concerning DCA (Fig. 6a), I wonder how the Authors traced the limits of the areas/groups and why a group of samples remained outside these groups, at the left edge of the biplot. I think that these points should be explained in text.

Author’s response: The bounding areas were drawn based on how the data naturally separated out and not any specific factoring program. 

We’ve checked those samples on the far left that weren’t included in the ACW box, and thank the reviewer for bringing this to our attention. We have edited the figure bounding box and included those samples, which are affiliated with Alaska Coastal water.

• I suggest to integrate the section “Time-series analysis” into the discussion section “Temporal trends in BCB Sea indicator species”.

Author’s response: We consider the Time-series analysis to be part of the Results so the interpretation and implications are discussed in broader terms in the Discussion section, “Temporal trends.”

• Why any environmental gradient has been identified along the transect “Offshore Hanna Shoal” and “Alaskan Beaufort Sea”? I think that these points deserve a little bit of discussion.

Author’s response: We thank the reviewer for pointing this out and have added these points about the Hanna Shoal and Beaufort Sea to the respective sections: Both of these areas lack strong gradients. Hanna Shoal includes samples farther offshore around the shoal where sea ice usually persists longer, so that Shoal may have an influence based upon sea ice persistence. The transects across the Beaufort Sea represented a fairly uniform water mass because it represents 20-100m water depth, except closer to river inputs where H. sorbyana increased in abundance, as noted in the discussion. Also, these two areas were limited in sampling years since they have not been sampled consistently as part of the DBO sampling program. The Hanna Shoal samples were collected in 2012-2013 and the Beaufort Sea compared primarily in 2018 and 1971 with limited intermittent sampling in years in between.

• Figure 1: I suggest to insert a map for an easier localization of the studied area.

Author’s response: We have added an inset map to Fig. 1 for context of the study area.

• I suggest to add the ostracod matrix used for multivariate analyses as supplementary material.

Author’s response: Yes we have added this as an Excel file table to the supplementary material (Supplement Table 1).

Reviewer #2: 

Dear Gemery and co-authors, your manuscript "Biogeography and ecology of Ostracoda in the U.S. northern Bering, Chukchi, and Beaufort Seas" is very well written and organised. I have annotated a pdf with some comments mainly about statistical analysis. 

Author’s response:

As you will see in the tracked manuscript copy, we have made all the changes suggested by the Reviewer and also included comments in the pdf attachment. We have also added several references that were not included originally but cited in the text.

I suggest to add a plate with the indicator taxa and to revise the reference list.

Author’s response: We have added SEM images of the fauna to the Supplement Table 1 Figure.

---

## [Editor Report · Decision Letter 1]

21 Apr 2021

Biogeography and ecology of Ostracoda in the U.S. northern Bering, Chukchi, and Beaufort Seas

PONE-D-21-04622R1

Dear Dr. Gemery,

We’re pleased to inform you that your manuscript has been judged scientifically suitable for publication and will be formally accepted for publication once it meets all outstanding technical requirements.

Kind regards,

Fabrizio Frontalini

Academic Editor

PLOS ONE
---

## [Editor Report · Acceptance letter]

26 Apr 2021

PONE-D-21-04622R1 

Biogeography and Ecology of Ostracoda in the U.S. northern Bering, Chukchi, and Beaufort Seas 

Dear Dr. Gemery:

I'm pleased to inform you that your manuscript has been deemed suitable for publication in PLOS ONE. Congratulations! Your manuscript is now with our production department. 

Kind regards, 

on behalf of

Dr. Fabrizio Frontalini 

Academic Editor

PLOS ONE